# Photooxidation of cyclohexene in the presence of SO$_2$: SOA yield and chemical composition

Shijie Liu[1,2,3], Long Jia[2], Yongfu Xu[2], Narcisse T. Tsona[1], Shuangshuang Ge[2], Lin Du[3,1,2]

[1]Environment Research Institute, Shandong University, Jinan, 250100, China
[2]State Key Laboratory of Atmospheric Boundary Layer Physics and Atmospheric Chemistry, Institute of Atmospheric Physics, Chinese Academy of Sciences, Beijing, 100029, China
[3]Shenzhen Research Institute, Shandong University, Shenzhen, 518057, China

*Correspondence to*: Lin Du (lindu@sdu.edu.cn); Yongfu Xu (xyf@mail.iap.ac.cn)

**Abstract.** Secondary organic aerosol (SOA) formation from cyclohexene/NOx system with various SO$_2$ concentrations under UV light was investigated to study the effects of cyclic alkenes on the atmospheric environment in polluted urban areas. A clear decrease at first and then increase of the SOA yield was found with increasing SO$_2$ concentrations. The lowest SOA yield was obtained when the initial SO$_2$ concentration was in the range of 30-40 ppb, while higher SOA yield compared to that without SO$_2$ could not be obtained until the initial SO$_2$ concentration was higher than 85 ppb. The decreasing SOA yield might be due to the fact that the promoting effect of acid-catalyzed reactions on SOA formation was less important than the inhibiting effect of decreasing OH concentration at low initial SO$_2$ concentrations, caused by the competition reactions of OH with SO$_2$ and cyclohexene. The competitive reaction was an important factor for SOA yield and it should not be neglected in photooxidation reactions. The composition of organic compounds in SOA was measured using several complementary techniques including Fourier transform infrared (FTIR) spectroscopy, ion chromatography (IC) and Exactive-Orbitrap mass spectrometer equipped with electro-spray interface (ESI). We present new evidence that organosulfates were produced from the photooxidation of cyclohexene in the presence of SO$_2$.

## 1 Introduction

Alkenes are widely emitted from biogenic and anthropogenic sources (Kesselmeier et al., 2002; Chin and Batterman, 2012), and their gas-phase oxidation reactions with OH, NO$_3$, or O$_3$ are among the most important processes in the atmosphere (Atkinson, 1997; Stewart et al., 2013; Paulson et al., 1999). Reactions of ozone with alkenes are an important source of free radicals in the lower atmosphere and thus, highly influence the oxidative capacity of the atmosphere (Paulson and Orlando, 1996). Some products of these reactions have sufficiently low vapor pressures to condense with other gaseous species, and contribute to the secondary organic aerosol (SOA) mass (Sarwar and Corsi, 2007; Sakamoto et al., 2013; Nah et al., 2016; Kroll and Seinfeld, 2008; Hallquist et al., 2009). SOA formation from the oxidation of VOCs has been receiving significant attention since recent years due to its large implication in the formation of atmospheric fine particulate matter (Jimenez et al., 2009). SOA has significant impacts on human health (Pope III and Dockery, 2006), air quality (Kanakidou et al., 2005; Jaoui

et al., 2012; McFiggans et al., 2006), and global climate change (Hansen and Sato, 2001; Adams et al., 2001; Pokhrel et al., 2016).

Although cyclic alkenes widely exist in the atmosphere, their gas-phase oxidation has received less attention than that of linear or branched alkenes (Sipilä et al., 2013). Cyclohexene is an important industrial chemical (Sun et al., 2013), and is also widespread in urban areas (Grosjean et al., 1978). Cyclohexene has been extensively studied as a monoterpene surrogate for inferring oxidation mechanisms and aerosol formation characteristics, because it has the basic structural unit in abundant biogenic monoterpenes and sesquiterpenes (Carlsson et al., 2012; Keywood et al., 2004b). The rate constants for gas-phase reactions of cyclohexene with OH, $O_3$ and $NO_3$ were measured at room temperature to be $(6.4\pm0.1)\times10^{-11}$, $(8.1\pm1.8)\times10^{-17}$ and $(5.4\pm0.2)\times10^{-13}$ $cm^3$ $molecule^{-1}$ $s^{-1}$, respectively (Stewart et al., 2013; Aschmann et al., 2012), and a correlation between the logarithm of the rate constants and the molecular orbital energies for simple cyclic alkenes was observed. The effect of pressure and that of the presence of $SO_2$ on the formation of stable gas-phase products and SOA from the ozonolysis of cyclohexene were investigated (Carlsson et al., 2012). It was found that the collisional stabilization of initial clusters was an important aspect for SOA formation processes involving sulfuric acid ($H_2SO_4$) and organic compounds. The effect of the structure of the hydrocarbon parent molecule on SOA formation was investigated for a series of cyclic alkenes and related compounds (Keywood et al., 2004b), and the SOA yield was found to be a function of the number of carbons present in the cyclic alkenes ring. The relative SOA yields from ozonolysis of cyclic alkenes can be quantitatively predicted from properties of the parent hydrocarbons, like the presence of a methyl group and an exocyclic double bond.

$SO_2$, one of the most important inorganic pollutants in urban areas, plays an important role on SOA formation (Wang et al., 2005; Lonsdale et al., 2012; Liu et al., 2016). Seasonal variations of $SO_2$ concentrations were found to be consistent with seasonal variations of $PM_{2.5}$ concentration (Cheng et al., 2015). Smog chamber simulations have indicated that $SO_2$ could enhance the formation of SOA from the oxidation of VOCs under acidic conditions by increasing aerosol acidity and ammonium sulfate aerosol formation (Edney et al., 2005; Liu et al., 2016; Attwood et al., 2014). Anthropogenic $SO_2$ emissions can impact new particle formation, and SOA composition (Lonsdale et al., 2012).

Despite the existence of organosulfates in ambient aerosols was first observed in 2005 (Romero and Oehme, 2005), proper identification of these aerosols was made two years later. In a series of laboratory chamber studies, it was shown that organosulfates present in ambient aerosols collected from various locations mostly originate from acid-catalyzed reactions of SOA formed from photooxidation of α-pinene and isoprene (Surratt et al., 2007). Recently, different kinds of organosulfates have been observed in SOA around the world, and organosulfates have been identified as a group of compounds that have an important contribution to the total amount of SOA in the atmosphere (Surratt et al., 2008; Froyd et al., 2010; Kristensen and Glasius, 2011; Tolocka and Turpin, 2012; Wang et al., 2015). Laboratory chamber studies showed that OH/NOx/$O_3$-initiated reactions of BVOCs, such as isoprene, α-pinene, β-pinene, and limonene with sulfates or sulfuric acid are the main processes for organosulfates formation (Surratt et al., 2007; Surratt et al., 2008; Hatch et al., 2011). Despite the well-recognized presence of organosulfates in SOA, their formation and transformation processes can be complex and varied, depending on the nature of the original organic compound involved. Extensive studies on their formation have been performed and several

mechanisms based on a variety of reactions have been proposed. Using nuclear resonance techniques, isoprene-derived epoxides formed during isoprene photooxidation reactions were found to be important intermediates for organonitrates and organosulfates formation via potential SOA reactions (Darer et al., 2011; Hu et al., 2011). The authors further found that organonitrates could easily be transformed to organosulfates during hydrolysis in the presence of sulfate. Some studies also showed that 2-methyl-3-buten-2-ol (MBO), due to its larger emissions than isoprene in some regions (Baker et al., 1999), is an important precursor for organosulfates and SOA in the atmosphere, through its reactions with OH under NO and aerosol acidity conditions, and from acid-catalyzed reactive uptake of MBO-based epoxides formed during MBO photooxidation (Mael et al., 2015; Zhang et al., 2012; Zhang et al., 2014). Organosulfates formation was also found from oxidation of hydroxyhydroperoxides (Riva et al., 2016) and from heterogeneous reactions of $SO_2$ with selected long-chain alkenes and unsaturated fatty acids (Passananti et al., 2016).

Reactions with sulfates or $H_2SO_4$ were the main formation processes of organosulfates. Qualitative analyses of organosulfates in SOA have been gaining more attention and development since recent years (Lin et al., 2012; Shalamzari et al., 2013; Staudt et al., 2014). Riva et al. investigated the formation of organosulfates from photooxidation of polycyclic aromatic hydrocarbons and found that, in the presence of sulfate aerosol, this photooxidation was a hitherto unrecognized source of anthropogenic secondary organosulfur compounds (Riva et al., 2015). A more complete structural characterization of polar organosulfates that originate from isoprene SOA was performed (Shalamzari et al., 2013), and an organosulfate related to methyl vinyl ketone and minor polar organosulfates related to croton aldehyde were identified. However, there is no report about the yield and chemical composition of SOA obtained from photooxidation of cyclohexene in the presence of $SO_2$.

In the present work, we investigated the yields and chemical composition of SOA during cyclohexene photooxidation under different $SO_2$ concentrations conditions. A better understanding of the magnitude and chemical composition of SOA from different $SO_2$ concentrations will contribute to more accurate SOA prediction from anthropogenic sources and give valuable information related to air pollution in urban environments.

## 2 Methods

### 2.1 Chamber description

The experiments were performed in a 400 L Teflon FEP film chamber (wall thickness 125 μm) at the Institute of Atmospheric Physics, Chinese Academy of Sciences, Beijing. The details of the chamber, including the experimental setup and analysis techniques have been described elsewhere (Du et al., 2007; Jia and Xu, 2014), and only a brief description is presented here. The reactor was surrounded by 12 black light lamps (GE F40BLB) with emission band centered at 365 nm, which were used to simulate the spectrum of the UV band in solar irradiation. Stainless steel was covered on the chamber interior walls to maximize and homogenize the interior light intensity. The effective light intensity near the ultraviolet region plays a decisive role in the formation of photochemical smog (Presto et al., 2005b). The effective light intensity of the

chamber was represented by the photolysis rate constant of $NO_2$. In our study, the average effective light intensity was determined to be 0.177 $min^{-1}$. Both inlet and outlet of the chamber were made of Teflon material. Atmospheric pressure was maintained in the chamber at all times. All experiments were performed at room temperature (307±2 K) under dry conditions (RH < 10%). The wall loss is the decrease of the concentration of reactive gas phase species caused by adsorption on the

inner wall of the reactor. Possible reaction of residual reactants and products on the inner wall with gas phase species is another important reason for wall loss. The wall loss can directly affect the quantitative evaluation of the photooxidation rate and SOA yield. A correct estimation of the wall loss is therefore necessary for a reliable analysis of the experimental results of SOA yield. In the present study, the wall loss of cyclohexene in the chamber could be neglected since no decrease in its concentration was observed. The wall loss of $O_3$, NOx, and $SO_2$ were first order because $\ln([X]_0/[X]_t)$ had a good correlation

with time ($R^2$=0.994, 0.944, 0.999 for $O_3$, NOx, and $SO_2$, respectively). The measured wall loss rate constants for $O_3$, NOx and $SO_2$ were $5.05\times10^{-6}$, $7.04\times10^{-6}$ and $6.39\times10^{-6}$ $s^{-1}$, respectively. The average value of the wall loss rate constant of particles was $4.7\times10^{-5}$ $s^{-1}$, and the measured particle concentrations in this study were corrected using the same method as Pathak et al. (Pathak et al., 2007). Typical profiles of the gas and particle phases are given in Figure S1 of the Supplementary material.

Prior to each experiment, the chamber was cleaned by purging with purified dry air for at least 8 h until residual hydrocarbons, $O_3$, NOx or particles could not be detected in the reactor. Known amounts of cyclohexene were injected into a 0.635 cm diameter Teflon FEP tube and dispensed into the chamber by purified dry air. Typical initial cyclohexene concentrations were 500 ppb. NOx was injected by a gas-tight syringe to make the mixing ratio of NOx in the reactor around 95 ppb during all experiments. The mixed concentration ratios of cyclohexene/NOx were in the range 4.4-6.9. SOA

formation experiments were carried out under UV irradiation in the presence of NOx to produce $O_3$ and OH radicals for cyclohexene oxidation. Although initial VOCs, NOx and average OH concentrations were different from typical urban conditions, efforts were made to maintain initial concentrations of the reactants as similar as possible to make sure the effect of $SO_2$ was the main reason for changes in the SOA yield. More details on the experimental conditions are shown in Table 1.

## 2.2 Gas and particle measurements

Ozone concentration in the reactor was measured using ozone analyzer (Model 49C, Thermo Electron Corporation, USA). A NO-$NO_2$-NOx analyzer (Model 42C, Thermo Electron Corporation, USA) was used to monitor the NOx concentration. Measurement of $SO_2$ concentration was made using a $SO_2$ analyzer (Model 43i-TLE, Thermo Electron Corporation, USA). The uncertainty of the $O_3$, NOx and $SO_2$ measurements was less than ±1%. The detection limits of the different monitors were 0.40 ppb, 0.50 ppb, and 0.05 ppb for NOx, $O_3$, and $SO_2$, respectively.

Two Tenax absorption tubes (150 mm length × 6 mm O.D., 0.2 g sorbent) were used to collect the sample before the UV lights were turned on and at the end of each experiment, respectively. The volume of the sample was 60 mL and the sampling time was 3 min. The concentrations of cyclohexene were analyzed by thermal desorption-gas chromatography-mass spectrometry (TD-GC-MS). A thermal desorption unit (Master TD, Dani, Italy) was combined with a 6890A gas

chromatograph (6890A, Agilent Tech., USA) interfaced to a 5975C mass selective detector (5975C, Agilent Tech., USA). The GC was equipped with a HP-5MS capillary column (30 m × 0.25 mm, 0.25 μm film thickness). The TD temperature was 280 °C, and the sampling time was 3 min. The GC-MS temperature program was as follows: the initial temperature of 40 °C was held for 4 min, and then raised to 300 °C at a rate of 20 °C min$^{-1}$. The inlet temperature was set at 250 °C and the transfer line at 200 °C. The ionization method in MS was electron impact ionization, and helium was used as the carrier gas at a constant flow (1.2 mL min$^{-1}$). Because a very diverse range of compounds might be present in the samples, the SCAN mode (36-500 amu) was used in the MS detector. This mode is known to be a classical and typical detection method for GC-MS analysis. The results were analyzed with MSD Productivity ChemStation.

Particle number concentrations and size distributions were measured with a scanning mobility particle sizer (SMPS), which consists of a differential mobility analyzer (DMA model 3081, TSI Inc., USA) and a condensation particle counter (CPC model 3776, TSI Inc., USA). A sheath flow/aerosol flow relationship of 3.0/0.3 L min$^{-1}$ was used for the measurements. The particle size was measured in the range of 14 to 710 nm, and each scan was 180 s. An aerosol density of 1.2 g cm$^{-3}$ was assumed to convert the particle volume concentration into the mass concentration (Zhang et al., 2015). Size distribution data were recorded and analyzed using the TSI AIM software v9.0.

### 2.3 SOA composition analysis

The chemical composition of SOA was important for analyzing the degree of cyclohexene oxidation, and it was used to evaluate the transformation from gas phase to particle phase. Particle phase chemical composition was studied by means of Fourier transform infrared (FTIR) spectroscopy (Nicolet iS10, Thermos Fisher, USA). Aerosols were sampled through a Dekati low pressure impactor (DLPI, DeKati Ltd, Finland). The impactor was connected to a pump working at a flow rate of 10 L min$^{-1}$ while sampling a total volume of 300 L of gas for each experimental run. Aerosols, from 108 to 650 nm diameter, were collected on an ungreased zinc selenide (ZnSe) disk (25 mm in diameter) for FTIR measurements.

The characteristic bands of inorganic and organic sulfates overlapped in the IR spectrum. In order to distinguish between these inorganic and organic sulfates, ion chromatograph (IC, Dionex ICS-900, Thermo Fisher, USA) was used to analyze the inorganic sulfate anion (SO$_4^{2-}$) in SOA. The limit of detection for the IC analysis was 0.005 μg mL$^{-1}$. SOA collected on ZnSe disks was firstly dissolved in high purity water (7 mL) and then measured by IC for SO$_4^{2-}$ concentrations. Anions were analyzed with a Dionex IonPac AS14A analytical column and an anion self-regenerating suppressor Dionex ASRS was used as eluent. The flow rate was 1.0 mL min$^{-1}$ with a mixture of 8.0 mmol L$^{-1}$ Na$_2$CO$_3$ and 1.0 mmol L$^{-1}$ NaHCO$_3$ for anions analyses. The suppressing current was 50 mA.

Chemical characterization of aerosols from photooxidation of cyclohexene was performed using an Exactive-Orbitrap mass spectrometer equipped with electro-spray interface (ESI) (Thermo Fisher Scientific, USA) operated in negative (-) ion mode, which was calibrated using the manufacturer's calibration standards mixture allowing for mass accuracies <5 ppm in external calibration mode. The capillary voltage was set to 3 kV. The desolvation gas flow was 200 μL min$^{-1}$, and the

desolvation gas temperature was 320 °C. SOA was collected on the aluminium foil using the same method as FTIR analysis and then extracted with 1 mL of acetonitrile. The aluminium foil was used due to its ease to handle and its non-reactivity with the sample. A total volume of 300 L was sampled at a flow rate of 10 L min$^{-1}$. A volume of 5 μL of the extraction and a direct injection were used for the measurement. Xcalibur 2.2 software (Thermo Fisher, USA) was used for the calculation of
chemical formulae from accurate measurement of $m/z$ values.

## 2.4 Chemicals

The chemicals used and their stated purities were as follows: cyclohexene (99%) was obtained from Aldrich and used without further purification. A zero air generator (Model 111, Thermo Scientific, USA) was used to generate clean air. The zero air has no detectable non-methane hydrocarbons (NMHC < 1 ppb), NOx (< 1 ppb), low O$_3$ concentration (< 3 ppb), low
particle numbers (< 5 cm$^{-3}$), and relative humidity (RH) below 10%. Ozone was produced from O$_2$ via electrical discharge using a dynamic gas calibrator (Model 146i, Thermo Scientific, USA). NO$_2$ (510 ppm), NO (50 ppm) and SO$_2$ (25 ppm) with ultra-pure N$_2$ (99.999%) as background gas was purchased from Beijing Huangyuan Gas Co., Ltd., China.

## 3 Results and discussion

### 3.1 Effect of SO$_2$ on SOA number concentrations

The particle number concentrations at the maximum SOA yield for the cyclohexene/NOx/SO$_2$ system with different initial SO$_2$ concentrations are shown in Figure 1, while the cooperation of the maximum number concentration and the particle number concentrations at the maximum SOA yield is shown in Figure S2. After the black light lamps were turned on, the SOA number concentrations increased rapidly to reach the maximum within 0.5 h in each experiment. Subsequently, the particle number concentrations gradually decreased accompanied by the growth of particle size by coagulation. The SOA
mass concentration kept increasing until its maximum was reached (after ~2 h). Both types of particle number concentrations had similar trends against initial SO$_2$ concentrations. In general, maximum particle number concentrations were three times higher than the particle number concentrations at the maximum SOA yield. In the remainder of this paper, in order to better elaborate the effect of SO$_2$ on the formation of particles, the particle number concentration refers to the particle number concentrations at the maximum SOA yield.
The particle number concentration increased with initial SO$_2$ concentration, and this increase could be divided into two stages: increasing stage and stable stage. In the increasing stage, with the initial SO$_2$ concentration increasing from 0 ppb to 30 ppb, the particle number concentration grew significantly under low initial SO$_2$ concentration (<5 ppb), then the growth rate reduced gradually. In the stable stage, when the SO$_2$ concentrations were varied systematically between 30 and 105 ppb, particle number concentrations were practically maintained steady, and there was no further obvious growth as shown in
Figure 1. For experiments with high initial SO$_2$ concentrations, the particle number concentrations were 10 times higher than those without SO$_2$, indicating enhanced new particle formation (NPF) when adding SO$_2$. It is evident from Figure 1 that even

small amounts of $SO_2$ affect the new particle formation substantially, as observed in previous studies (Chu et al., 2016; Liu et al., 2016).

Nucleation is a fundamental step in the atmospheric new particle formation. Nucleation of particles in the atmosphere has been observed to be strongly dependent on the abundance of $H_2SO_4$ (Sihto et al., 2006; Xiao et al., 2015). Normally, $SO_2$ was deemed to be oxidized by OH radicals to form $H_2SO_4$ through homogeneous reactions in gas phase (Calvert et al., 1978), or by $H_2O_2$ and $O_3$ through in-cloud processes in aqueous phase (Lelieveld and Heintzenberg, 1992). However, the aqueous phase formation of $H_2SO_4$ is negligible in this study (RH<10%). As the precursor of $H_2SO_4$, $SO_2$ at high concentrations would lead to more $H_2SO_4$ formation, and thereby increase the nucleation rates and total particle number concentrations (Sipilä et al., 2010). Because of the similar initial conditions for each experiment except $SO_2$, the amount of OH radicals produced was assumed to be almost equal. In the presence of high concentrations of $SO_2$, new particle formation was not enhanced. This feature may indicate that no more sulfates were formed when $SO_2$ was in large excess (>30 ppb) and the OH radicals were insufficient. The quantity of OH radicals was the main restraint on $H_2SO_4$ formation at high initial $SO_2$ concentrations in the present study. Therefore, the particle number concentration was maintained steady and was independent of the $SO_2$ concentrations in the second stage.

Besides, the mean diameter of particles increased with photooxidation reaction time, which suggests that only few particles were generated after a burst increase at the initial stage of SOA formation. Once new particles are formed, there is a competition between growth of existing particles by uptake of the precursors and formation of new particles. Our result agrees with previous studies that there was no obvious increase in aerosol number concentration when additional VOCs were injected, but a significant increase in SOA mass concentration (Presto et al., 2005b). As long as there was enough seed particle surface area, vapor condensation onto existing aerosol particles was favored compared to the formation of new particles, and this condensation would be the main contribution to the increase of SOA mass.

## 3.2 Effect of $SO_2$ on SOA yields

SOA yield (Y) is defined as $Y=\Delta M_0/\Delta HC$, where $\Delta M_0$ is the produced organic aerosol mass concentration ($\mu g\ m^{-3}$), and $\Delta HC$ is the mass concentration of reacted cyclohexene ($\mu g\ m^{-3}$). The SOA yields of cyclohexene at different $SO_2$ concentration as determined by SMPS are shown in Figure 2. The numerical values of the aerosol mass concentration and SOA yields at different conditions are shown in Table 1.

The SOA yields in the absence of $SO_2$ were in the range of 2.7-3.4%, which were an order of magnitude lower than those reported in previous studies (Warren et al., 2009; Keywood et al., 2004b; Kalberer et al., 2000). There are three possible explanations to this phenomenon. (1) SOA formation is closely related to the oxidation capacity in the photooxidation experiments and, therefore, is affected by the ratio of $[VOC]_0/[NOx]_0$ (Pandis et al., 1991). Experiments performed with different $SO_2$ concentrations indicate that the SOA formation is partly controlled by the ability of the system to oxidize cyclohexene and contribute to the particle mass. As indicated in Figure S3 of the Supplementary material, even at 0 ppb of $SO_2$, the mass concentration of SOA quickly reaches its maximum. Experiments with higher NOx levels have been proved to

get considerably lower SOA yield than those with lower NOx levels at the same VOCs concentration (Song et al., 2005). Reactions of organo-peroxy radicals ($RO_2$) with NO and $NO_2$ instead of peroxy radicals ($RO_2$ or $HO_2$) under high NOx conditions resulted in the formation of volatile organic products and a decreased SOA yield (Lane et al., 2008). It was reported that SOA yield was constant for $[VOC]_0/[NOx]_0>15$, but decreased considerably (by a factor of more than 4) as

$[VOC]_0/[NOx]_0$ decreased (Presto et al., 2005a). In this study, the $[VOC]_0/[NOx]_0$ ratio was maintained at about 4.4 to 6.9. Recently, the NOx dependence of SOA formation from photooxidation of β-pinene was comprehensively investigated (Sarrafzadeh et al., 2016), and it was shown that the NOx-induced OH concentration was the major factor influencing the SOA yield. The impacts of NOx on SOA formation were only moderate if the impact of NOx on OH concentration was eliminated. The OH concentration in our study was relatively insufficient, which was the main limiting factor for SOA

formation. (2) UV light is another factor influencing the SOA yield. SOA yields between dark and UV-illuminated conditions were reported to be different (Presto et al., 2005b). Exposure to UV light could reduce SOA yield by 20-40%, while more volatile products were formed (Griffin et al., 1999). (3) The temperature may have a pronounced influence on SOA yield (Qi et al., 2010; Emanuelsson et al., 2013). At low temperatures, semi-volatile organic compounds would favor the condensation of gas phase species and a higher SOA yield could be expected. Raising the chamber temperature by 10 K

should cause a decrease of 10% in aerosol yield (Pathak et al., 2007). SOA yields reported in the present study were obtained at a higher temperature (307±2 K) than 298 K used in most previous studies. On the basis of the discussion above, the SOA yield from cyclohexene in this study was lower than observed in the previous studies.

SOA yields for the cyclohexene/NOx/$SO_2$ system were measured for initial $SO_2$ mixing ratios of 0-105 ppb. Due to the error associated with the $SO_2$ concentrations measurement, with stronger impact on low values than on higher values, several

experiments were performed at $SO_2$ concentrations below 40 ppb. The experimental results showed a clear decrease at first and then an increase in the SOA yield with increasing $SO_2$ concentrations (Figure 2). When $SO_2$ concentrations increased from 0 to 40.8 ppb, there was a remarkable decrease in SOA yield, dropping by about half with the increase of $SO_2$ concentration. For $SO_2$ concentrations higher than 40.8 ppb, the SOA yield increased with increasing $SO_2$ concentration. The highest SOA yield was found to be 3.5%, and was at 104.7 ppb $SO_2$ concentration. The lowest SOA yield of cyclohexene

photooxidation was obtained at an initial $SO_2$ concentration of 40 ppb. Although the SOA yield increased gradually with the initial $SO_2$ concentration at concentrations higher than 40 ppb, a higher SOA yield than that in the absence of $SO_2$ could not be obtained when the initial $SO_2$ concentration was lower than 85 ppb.

Both NO and $NO_2$ were used as NOx for repeated experiments in the current study. Although the photooxidation reaction could not happen in the case of NO until it was oxidized to $NO_2$, which means that both NO- and $NO_2$-initiated

photooxidation reactions were actually triggered by $NO_2$, the chemistry of SOA formation from both processes is similar. Despite the time of occurrence of the maximum SOA concentration for the experiment with $NO_2$ was half an hour earlier than that for the experiment with NO, the results of SOA yield were similar.

In the presence of $SO_2$, enhanced SOA formation could be attributed to acid-catalyzed heterogeneous reactions (Jang et al., 2002; Xu et al., 2014). When studying the effect on acidic seed of the growth of isoprene- and α-pinene-based SOA, it was

shown that FTIR peaks at 1180 cm$^{-1}$ (C-O-C stretch of hemiacetals and acetals) and 1050 cm$^{-1}$ (C-C-O asymmetric stretch of alcohols) are indicators of acid-catalyzed heterogeneous reactions since these peaks could not, otherwise, be observed in non-acidic conditions (Jang et al., 2002; Czoschke et al., 2003). These peaks are prominent in IR spectra from SOA formation in an acidic particle environment. In the current study, similar peaks were observed at 1195 and 1040 cm$^{-1}$ (see Figure S4). Their intensities were very weak when initial SO$_2$ concentrations were lower than 44 ppb, indicating that acid-catalyzed reactions were not facilitated at these conditions.

However, there were some undiscovered processes that could inhibit the formation of SOA in the cyclohexene/NOx/SO$_2$ system. The competitive reaction between SO$_2$ and cyclohexene might be among the reasons for the decrease in the SOA yield. For example, SO$_2$ could be oxidized by OH to form H$_2$SO$_4$ (Somnitz, 2004). Due to the presence of O$_3$ in our system, the formation of Criegee intermediates (CI) and their reactions with SO$_2$ could equally be expected (Criegee, 1975). The rate constants of O$_3$ + cyclohexene and OH + cyclohexene reactions were determined to be $7.44 \times 10^{-17}$ and $6.09 \times 10^{-11}$ cm$^3$ molecule$^{-1}$ s$^{-1}$, corresponding respectively to 5.5 h and 2.5 h lifetimes for cyclohexene (Treacy et al., 1997; Rogers, 1989). Hence, it is likely that the cyclohexene reaction with O$_3$ would be less important than the reaction with OH in this study. However, the importance of SO$_2$ reactions with stabilized could be limited due to the kinetics and low yield of the latter (Stewart et al., 2013; Keywood et al., 2004a; Hatakeyama et al., 1984). As mentioned above, the photooxidation in this study was at high-NOx conditions and the OH was the main limiting factor for SOA formation because of its relatively low concentration. The change of cyclohexene concentration with time at different initial SO$_2$ concentrations is shown in Figure S5. It is seen that in the first half hour, the amount of cyclohexene consumed is almost similar for different SO$_2$ concentrations. Regarding the difference between initial cyclohexene concentrations, the similar amount of reacted cyclohexene in the first half hour indicates that low and high OH concentrations were used at high and low SO$_2$ conditions, respectively. The consuming rate of cyclohexene was slightly higher without SO$_2$ in the chamber, which means that if there was a competition reaction, its effect was very limited. At lower OH concentration condition caused by the reaction between SO$_2$ and OH, the formation of SOA was inhibited.

The rate constant for the OH + SO$_2$ reaction was estimated to be $2.01 \times 10^{-12}$ cm$^3$ molecule$^{-1}$ s$^{-1}$, corresponding to a SO$_2$ lifetime of 69 h (Atkinson et al., 1997). This reaction is much slower than the cyclohexene + OH reaction, suggesting that OH + SO$_2$ reaction has very little impact on the OH concentration in the system. In our experiment, the decrease in the SOA yield with SO$_2$ addition might then not be attributed to its reaction with OH. It is also possible that the SO$_2$ addition could change the chemistry of the photooxidation process and suppress the oxygenation of products (Friedman et al., 2016; Liu et al., 2015). Comparing the MS results at different initial SO$_2$ concentrations, the proportion of low molecular weight components increases with increasing SO$_2$ concentration. Molecular weights have negative correlation with volatility, which could also make the SOA yield to decrease. Moreover, in real atmospheric situations where O$_3$ is found in much higher proportion than OH, cyclohexene would mainly react with O$_3$ to produce Criegee intermediates, which are good SO$_2$ oxidizers, and significantly less SOA than in the chamber would be formed. Accordingly, SOA yield showed descending trend with the increase of SO$_2$ concentrations when they are below 40 ppb.

When the initial $SO_2$ concentration was greater than 40 ppb, the acid-catalyzed heterogeneous formation of SOA became more significant (Figure 2). The same SOA yield was obtained in the absence of $SO_2$ and at 85 ppb initial $SO_2$ concentration. The competitive reaction plays an important role on SOA formation, and it should be taken into account in SOA simulation models or air quality models for more accurate predictions. Acid-catalyzed reactions gradually became important as the initial $SO_2$ concentration for SOA yield increased. The formation of low volatile organics (e.g. organosulfates) by photooxidation in the presence of $SO_2$ might be another reason for the increase of the SOA yield.

### 3.3 Organosulfates formation

When $SO_2$ was added into the chamber, acidic aerosol particles were formed by photooxidation of $SO_2$ in a reaction initiated by OH. The amount of $SO_4^{2-}$ in particle phase and the consumption of $SO_2$ ($\Delta SO_2$) with varying initial $SO_2$ concentrations are shown in Figure 3. The changes with initial $SO_2$ concentrations were not uniform between the $SO_4^{2-}$ concentration and $\Delta SO_2$, which indicates that besides $SO_4^{2-}$, other products were formed from the reaction of $SO_2$. Typical IR spectra of aerosols from the cyclohexene/NOx/$SO_2$ system under different $SO_2$ concentrations are presented in Figure 4. Based on the peak positions in the IR spectra, different functional groups were assigned. The broadband at 3100 to 3300 cm$^{-1}$ is assigned to the O–H stretching of hydroxyl and carboxyl groups (Coury and Dillner, 2008), while the peak at 1717 cm$^{-1}$ represents the C=O stretching of aldehydes, ketones, and carboxylic acids. The peaks at 1622 and 1278 cm$^{-1}$ show good correlation and both are assigned to the -$ONO_2$ stretching (Liu et al., 2012; Jia and Xu, 2014). The characteristic absorption band at 1500-1350 cm$^{-1}$ is the C–O stretching and O–H bending of the COOH group (Ofner et al., 2011), and the absorption peak of sulfate exists in the range of 1200-1000 cm$^{-1}$ (Wu et al., 2013). The band at 1100 cm$^{-1}$ in the IR spectra can be attributed to the sulfate group in organic compounds and sulfate. It has been confirmed that the S=O absorption band in organic sulfate monoesters appears around 1040-1070 cm$^{-1}$ (Chihara, 1958). Although, more studies on bands assignments in organosulfates are not currently available from the literature for further comparison, the 1100 cm$^{-1}$ band from the current FTIR study can reasonably be assigned to S=O in the sulfate group.

The intensities of most absorption bands, such as O-H at 3100-3300 cm$^{-1}$, C=O at 1717 cm$^{-1}$, -$ONO_2$ at 1622 and 1278 cm$^{-1}$, and C-H at 2930 cm$^{-1}$, have similar trends with the change of SOA yield for initial $SO_2$ concentrations between 11 and 105 ppb. However, the band of sulfate at 1100 cm$^{-1}$ in IR spectra increases with the rise of initial $SO_2$ concentration rather than the SOA yield, which suggests the formation of sulfate group in organic compounds and sulfate product from $SO_2$ photooxidation since, only the relative difference in the intensities of FTIR peaks were studied here. The relative intensity of the band at 1100 cm$^{-1}$ increased by 1.8 times when the initial $SO_2$ concentration rose from 0 to 44 ppb, and increased by 7.2 times when the initial $SO_2$ concentration was 105 ppb. This intensity band grew slowly at low $SO_2$ concentrations due to the decrease in the formation of aerosols. To clearly show the amount of sulfate group and sulfate in aerosols, the intensity of the band at 1100 cm$^{-1}$ and the amount of $SO_4^{2-}$ were compared in the same aerosol mass, as shown in Figure 5. The relative intensity was set to 1 when the initial $SO_2$ concentration was 44.3 ppb.

The relative intensities of the band at 1100 cm$^{-1}$, which represented the intensity of both SO$_4^{2-}$ and the sulfate group in organic compounds, increased approximately in a linear form with the increase of initial SO$_2$ concentration (R$^2$=0.91). If the 1100 cm$^{-1}$ band originated solely from SO$_4^{2-}$, the change of the band intensity would be consistent with SO$_4^{2-}$ concentration in unit mass of aerosols. Figure 5 shows the inconsistency between the trends of FTIR band at 1100 cm$^{-1}$ and the amount SO$_4^{2-}$ as the initial SO$_2$ concentration increases, which implies that the 1100 cm$^{-1}$ band originated not only from SO$_4^{2-}$, but also from other organosulfur compounds. These include organosulfates, which also have the S=O bond, and might therefore contribute to the 1100 cm$^{-1}$ band in the FTIR spectrum. The difference between the trends of FTIR band at 1100 cm$^{-1}$ and the amount of SO$_4^{2-}$ with increasing initial SO$_2$ concentration can be attributed to the formation of organosulfates.

The composition of cyclohexene SOA was examined with Exactive-Orbitrap MS using negative ion mode ESI and the mass spectrum was recorded at a resolution of 10$^5$ (Figure 6). The OH addition to the C=C bond produces an alkyl peroxyl (RO$_2$) radical that can react with NO to yield organonitrates (Perring et al., 2013). Although the formation of organonitrates was highly expected, there was no evidence of the presence of N-containing compounds from the main peaks of Figure 6, indicating that organonitrates would be formed at very low concentrations, if at all. A similar conclusion could be observed from Figure 4, when noticing that the -ONO$_2$ stretching peaks at 1622 and 1230 cm$^{-1}$ have very low intensities. The presumed low concentrations of organonitrates might be due to the low concentration of NO when SOA was formed. RO$_2$ radicals also react with NO$_2$ to form peroxy nitrates (RO$_2$NO$_2$) on time scales comparable to RONO$_2$ formation. However, RO$_2$NO$_2$ are thermally labile and rapidly dissociate at ambient temperatures (Perring et al., 2013). Organosulfates were identified in the particle phase from the chamber experiments. Accurate mass fittings for measured ions of organosulfates in ESI negative ion mode are given in Table 2. As shown in Figure 6 and Table 2, 10 different organosulfates were successfully detected and identified from cyclohexene SOA. These results not only first prove the formation of organosulfates from cyclohexene photooxidation at high-NOx condition in the presence of SO$_2$, but also provide evidence and reference for organosulfates identification by FTIR-IC joint technique. A deprotonated molecular ion at $m/z$ = 195.03322 (C$_6$H$_{11}$O$_5$S$^-$) had the maximum content (more than 60%) of all the organosulfates detected in this study. Its intensity was 6.5 times higher than that of the second highest abundant organosulfate. The intermediate product of cyclohexene + OH reaction, i.e., CH(O)CH$_2$CH$_2$CH$_2$CH$_2$ĊHOH, has a hydroxyl group, and the organosulfate product ($m/z$=195.03322) would likely form from the intermediate product, not from the end product. This organosulfate, together with organosulfates with $m/z$ = 179.00181 and 209.01257 measured in this study were also measured in the Arctic sites, however, with unknown sources (Hansen et al., 2014). This study further supports the formation of organosulfates from cyclohexene in the atmosphere.

The mass spectra show a great abundance of peaks, detected as deprotonated molecular ions (M−H)$^-$ formed via proton abstraction. Most cyclohexene SOA contained carboxylic acid and/or aldehyde moieties. The products of the reaction of OH radicals with cyclohexene in the presence of NO were investigated and were identified as cyclic 1,2-hydroxynitrates and 1,6-hexanedial (Aschmann et al., 2012). These products could not be detected by Exactive-Orbitrap MS in our study. Aldehydes could be oxidized by OH radicals to form carboxyls, which have been intensively identified in previous studies (Cameron et al., 2002; Goldsmith et al., 2012). 1,6-hexanedial might be further oxidized in the atmospheric photooxidation reactions to

form 1,6-adipic acid ($C_6H_{10}O_4$) and 6-oxohexanoic acid ($C_6H_{10}O_3$), which were both observed in this study. In addition to the $C_6$ compounds observed in this study, a $C_5H_7O_3^-$ ion was detected with higher abundance than the $C_6$ compounds. Although the formation of $C_5H_7O_3^-$ might be due to a carbonyl cleavage from a six-carbon atoms chain, a proper mechanism for its formation could not be determined. A $C_4$ compound was also detected likely as a result of a carbonyl cleavage from a $C_5$ compound. However, there was no evidence of the formation of compounds with less than four carbon atoms.

The Exactive-Orbitrap MS spectra of species formed from different initial $SO_2$ concentrations are shown in Figure S6. We found no obvious difference in the composition and response of organosulfates with different initial $SO_2$ concentrations. The relative intensity of the peak at $m/z$ = 97, which corresponds to sulfate, was set to 100% in both Exactive-Orbitrap MS spectra. The relative intensities of organosulfate peaks in both spectra were almost unchanged regardless of the initial $SO_2$ concentration. However, Minerath et al. and Hatch et al. observed an increase in organosulfate yields with increasing sulfate concentration, and sulfate can be regarded as a key parameter influencing the formation of organosulfates (Minerath and Elrod, 2009; Hatch et al., 2011). Since sulfate is formed as a result of $SO_2$ oxidation in the current study, quantification of organosulfates formed from cyclohexene photooxidation will be investigated in further studies in order to examine the effect of increasing $SO_2$ concentration on organosulfates formation. Comparing Exactive-Orbitrap MS data when $SO_2$ initial concentrations were 0 ppb and 236 ppb reveals that the bands representing organosulfates do not appear at 0 ppb of $SO_2$. Peaks at $m/z$ larger than 150 were undetectable at initial $SO_2$ concentration of 0 ppb, while products without sulfur peaked at both concentrations, with the only difference being their relative intensities. This implies that the process of SOA formation strongly depends on initial $SO_2$ concentrations.

**4 Conclusion**

We report a series of laboratory chamber studies on the formation of SOA from the mixture of cyclohexene and $SO_2$. The experiments were based on Fourier transform infrared spectroscopy, ion chromatography and electrospray ionization high-resolution quadrupole mass spectrometry, and were performed under NOx conditions. Although new particle formation was found to be enhanced with increasing $SO_2$ concentration, the yield of SOA was not enhanced for all $SO_2$ concentrations between 0 and 105 ppb. SOA formation decreased at first and then was enhanced for all $SO_2$ concentration above 40 ppb. Both acid-catalysis and competitive OH reactions with cyclohexene and $SO_2$ were found to have important effects on the SOA formation and hence, should be taken into account in SOA simulation models or air quality models for a better understanding of haze pollution. The formation of organosulfates, an important part of atmospheric organic aerosol components, was first observed from cyclohexene SOA. However, quantification of these organosulfates and precursors to their formation should be determined in further studies. The formation of organosulfates has a great significance for the particulate matter formation under high $SO_2$ concentrations in the atmosphere.

**Acknowledgments**

This work was supported by National Natural Science Foundation of China (91644214, 21577080, 41375129), Shenzhen Science and Technology Research and Development Funds, China (JCYJ20150402105524052), and the "Strategic Priority Research Program (B)" of the Chinese Academy of Sciences (XDB05010104).

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

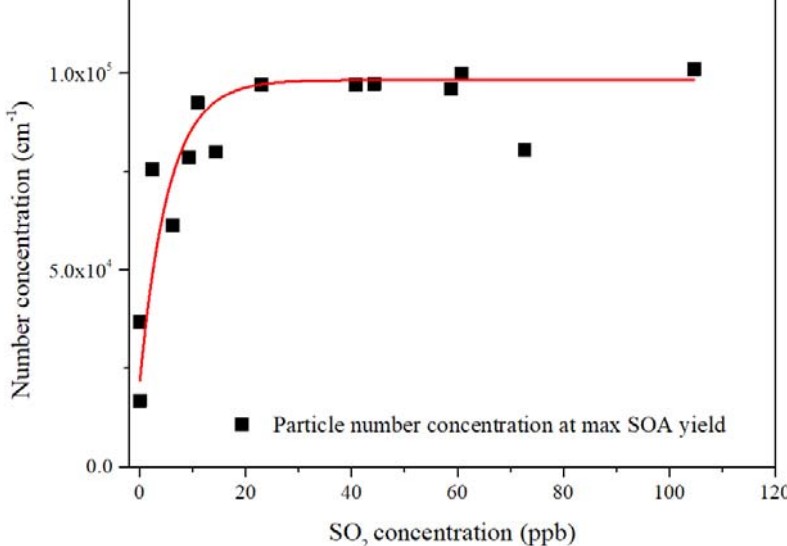

**Figure 1: Particle number concentrations of SOA in the photooxidation of the cyclohexene/NOx/SO₂ system with different initial SO₂ concentrations.**

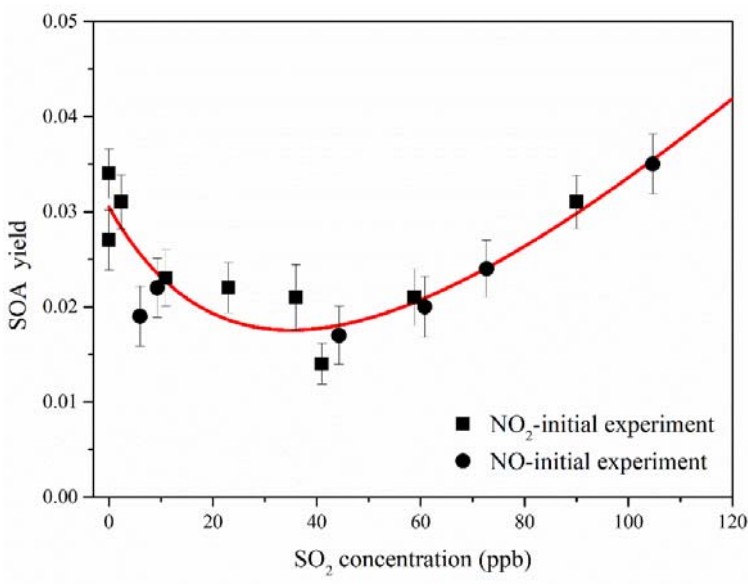

**Figure 2: SOA yields of cyclohexene photooxidation in the presence of NOx at different initial SO₂ concentrations. Solid line is the least-square fitting to the data. The error bars were determined on the basis of propagation of uncertainties arising in the ΔHC measurements, including GC calibration uncertainties propagation and the variance in the initial cyclohexene measurements.**

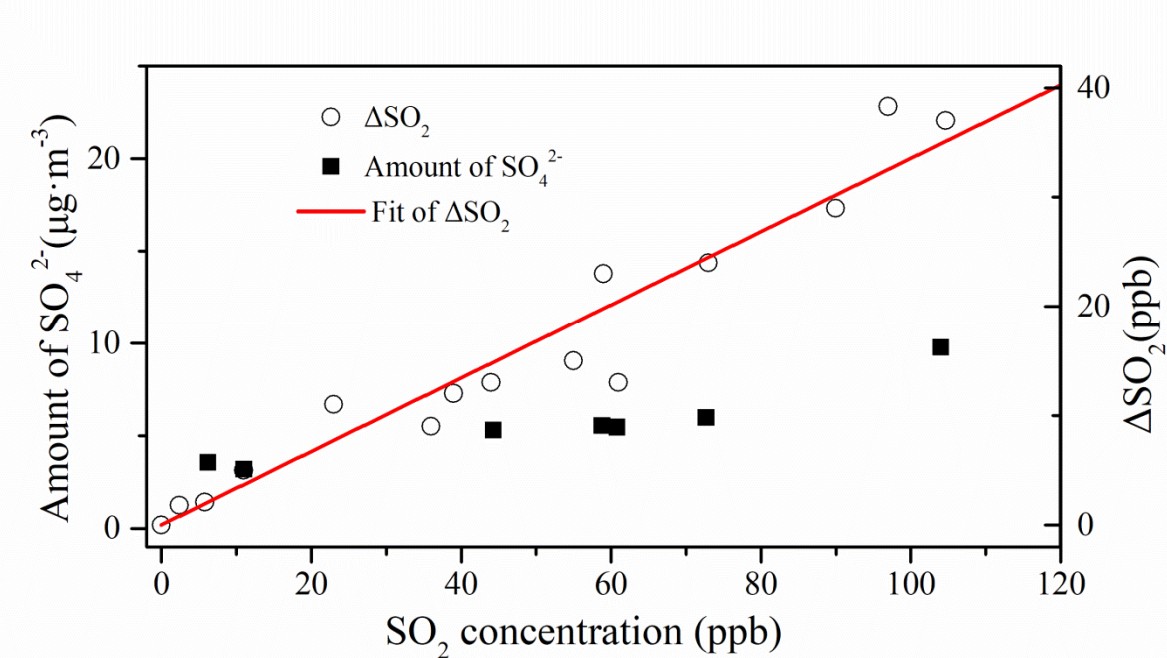

**Figure 3: The amount of SO$_4^{2-}$ in particle phase and the consumption of SO$_2$ (ΔSO$_2$) with different initial SO$_2$ concentrations.**

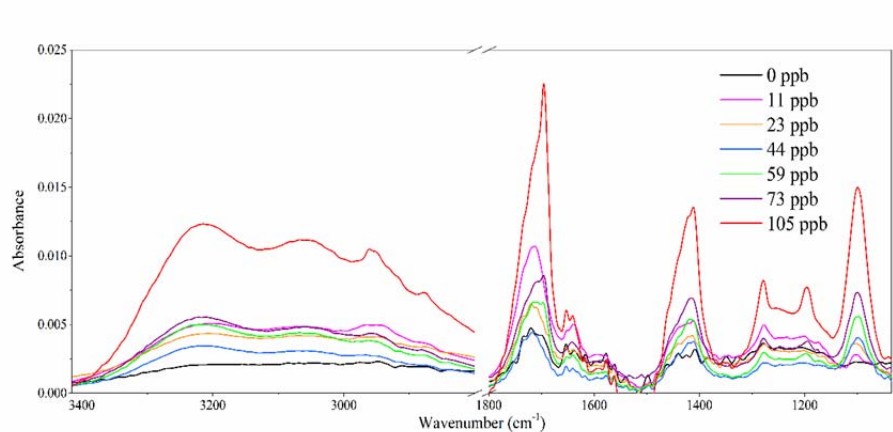

**Figure 4: IR spectra of aerosols from the cyclohexene/NOx/SO$_2$ system under different SO$_2$ concentrations.**

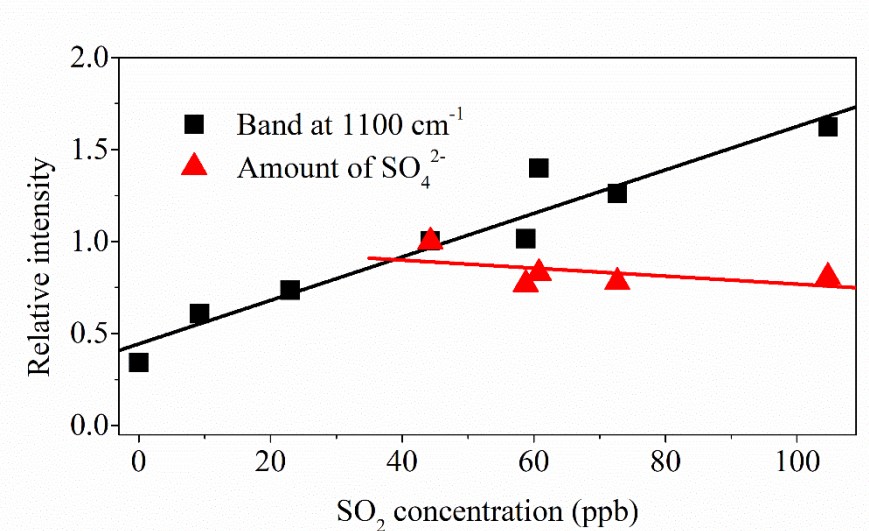

**Figure 5: The relative intensity of the FTIR band at 1100 cm$^{-1}$ (square) and the amount of SO$_4^{2-}$ (triangle) normalized to SOA mass. The 1100 cm$^{-1}$ band intensity and the amount of SO$_4^{2-}$ were divided by the formed SOA mass, firstly. Subsequently, the results of both FTIR band at 1100 cm$^{-1}$ and the amount of SO$_4^{2-}$ divided by SOA mass were set to 1 at the experiment which initial SO$_2$ concentration was 44.3 ppb.**

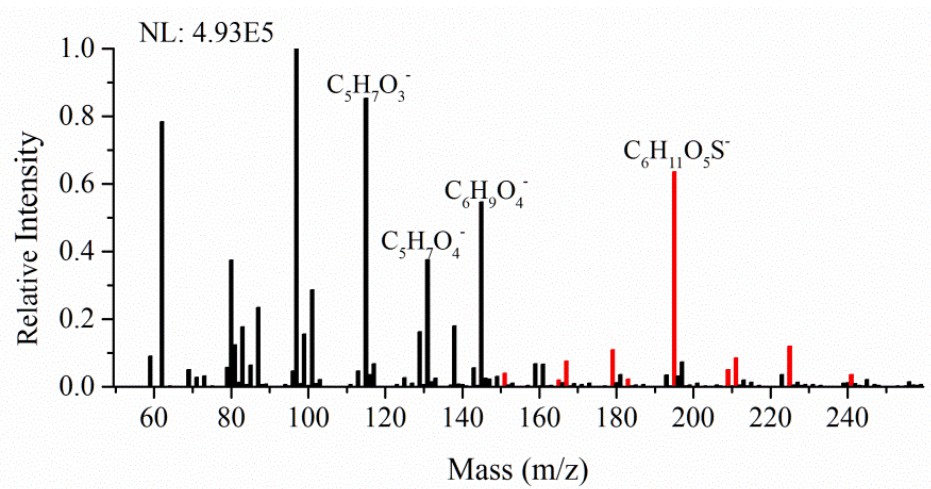

**Figure 6: Negative ion mode ESI mass spectrum of SOA generated from the photooxidation of cyclohexene in the presence of SO$_2$. Red peaks correspond to organic compounds containing the sulfate group. The mass resolution is 10$^5$.**

**Table 1 Experimental conditions for the photooxidation of cyclohexene/NOx/SO$_2$ system. All experiments were performed under dry conditions (relative humidity < 10 %). ΔM$_0$ is the produced organic aerosol mass concentration and Y is the SOA yield.**

| Exp. | T (K) | SO$_2$ (ppb) | cyclohexene (ppb) | NOx (ppb) | cyclohexene/NOx | ΔM$_0$ ($\mu$g m$^{-3}$) | Y (%) |
|------|-------|--------------|-------------------|-----------|-----------------|--------------------------|-------|
| 1[b] | 308 | 0.0 | 596 | 122.0 | 4.9 | 57.0 | 2.66 |
| 2[b] | 305 | 0.0 | 651 | 93.7 | 6.9 | 79.7 | 3.40 |
| 3[b] | 309 | 2.4 | 553 | 95.7 | 5.8 | 62.6 | 3.15 |
| 4[a] | 307 | 5.8 | 612 | 92.7 | 6.6 | 41.0 | 1.87 |
| 5[a] | 309 | 9.3 | 599 | 93.5 | 6.4 | 48.1 | 2.23 |
| 6[b] | 309 | 11.0 | 574 | 94.7 | 6.1 | 47.1 | 2.28 |
| 7[b] | 309 | 23.0 | 514 | 90.5 | 5.7 | 42.6 | 2.30 |
| 8[b] | 305 | 36.6 | 665 | 99.7 | 6.7 | 96.3 | 2.01 |
| 9[b] | 308 | 40.8 | 472 | 91.4 | 5.2 | 22.6 | 1.33 |
| 10[a] | 308 | 44.3 | 592 | 98.6 | 6.0 | 35.3 | 1.66 |
| 11[b] | 305 | 55.0 | 497 | 113.0 | 4.4 | 77.3 | 2.16 |
| 12[b] | 308 | 58.8 | 577 | 96.7 | 6.0 | 44.3 | 2.13 |
| 13[a] | 309 | 60.8 | 626 | 102.0 | 6.1 | 43.9 | 1.95 |
| 14[a] | 308 | 72.7 | 581 | 98.4 | 5.9 | 49.2 | 2.35 |
| 15[b] | 306 | 90.0 | 543 | 99.6 | 5.4 | 102.0 | 2.62 |
| 16[a] | 309 | 104.7 | 608 | 93.7 | 6.5 | 77.1 | 3.52 |
| 17[bc] | 305 | 236.0 | 1048 | 198.0 | 5.3 | - | - |
| 18[bc] | 306 | 93.7 | 1235 | 215 | 5.7 | - | - |

[a]: the experiment was initiated by NO.

[b]: the experiment was initiated by NO$_2$.

[c]: the formed particles were detected by Exactive-Orbitrap MS.

**Table 2. Accurate mass fittings for main products and measured organosulfates ions in ESI negative ion mode from cyclohexene photooxidation in the presence of SO₂ under high-NOx conditions**

| Measured [a] $m/z$ | Ion | Proposed Ion Formula | Delta [b] (ppm) | RDB [c] |
|---|---|---|---|---|
| 115.03942 | | $C_5H_7O_3^-$ | -5.628 | 2 |
| 145.05019 | | $C_6H_9O_4^-$ | -3.048 | 2 |
| 131.03444 | | $C_5H_7O_4^-$ | -4.136 | 2 |
| 101.06006 | $(M-H)^-$ | $C_5H_9O_2^-$ | -7.351 | 1 |
| 87.04433 | | $C_4H_7O_2^-$ | -9.453 | 1 |
| 129.05515 | | $C_6H_9O_3^-$ | -4.397 | 2 |
| 99.04439 | | $C_5H_7O_2^-$ | -7.702 | 2 |
| Organosulfates | | | | |
| 195.03322 | | $C_6H_{11}O_5S^-$ | -0.243 | 1 |
| 225.00771 | | $C_6H_9O_7S^-$ | 1.171 | 2 |
| 179.00181 | | $C_5H_7O_5S^-$ | -0.0879 | 2 |
| 211.02828 | | $C_6H_{11}O_6S^-$ | 0.464 | 1 |
| 167.00167 | $(M-H)^-$ | $C_4H_7O_5S^-$ | -1.780 | 1 |
| 209.01257 | | $C_6H_9O_6S^-$ | 0.182 | 2 |
| 151.00658 | | $C_4H_7O_4S^-$ | -3.130 | 1 |
| 241.00278 | | $C_6H_9O_8S^-$ | 1.738 | 2 |
| 182.99667 | | $C_4H_7O_6S^-$ | -1.158 | 1 |
| 164.98594 | | $C_4H_5O_5S^-$ | -2.287 | 2 |

[a] Sort by abundance intensity.

[b] Delta: label the peak with the difference between the theoretical and measured $m/z$.

[c] RDB: ring and double bond equivalent.