# Peer review of "Photooxidation of cyclohexene in the presence of SO2: SOA yield and chemical composition"

_Atmospheric Chemistry and Physics, 2017_

## Referee Comment (RC1) · Anonymous Referee #4 · 16 Mar 2017

This manuscript presents interesting new results on atmospheric reactions of photooxidation of cyclohexene in the presence of SO2. Unfortunately the quality of the presentation is not suitable for publication in ACP regarding scientific discussion and interpretation of results. The manuscript must be rewritten to discuss the results from a more objective, scientific point, which to a higher degree takes data uncertainty into account before reaching conclusions. Furthermore, there are many grammatical errors.

In the following I have explained some of the major points.

Introduction The introduction should include more references to relevant previous work. One example is in line 23 page 1, where only one study (on measurement methods for VOC from vehicles) is used as reference for the general statement about emission of alkenes from biogenic and anthropogenic sources. The reference Jimenez et al.

[Figure]

(page 1 line 28) on reactions of polyfluorinated compounds is not relevant for a general statement on SOA formation in the atmosphere, and it should thus be removed. On page 2 (lines 27-30) it is stated that a substantial amount of organosulfates have been observed. Most measurements estimate up to 10% of aerosol mass, and typically much lower in most places, which in my opinion is not a "substantial amount". Why so much focus on organosulfate formation from MBO, which is typically not found in high concentrations? Page 2 lines 11-21: This section has a good number of relevant references.

Experimental Page 4 line 8: The VOC/NOx ratio was not about 5, but varied from 4.4 to 7. Section 2.3: Why were only aerosols in the range 108-650 nm collected? Were samples for FTIR and SOA analysis collected right after each other? What was the limit of detection of the IC analysis?

Results and discussion Section 3.1 should be moved to the experimental section.

Section 3.2 Page 6 line 22. The trend is not clear, especially regarding maximum particle number, which shows quite some scatter. Page 6 line 26-32: The conclusions in this section are beyond what I see in the data, given the scatter. Why are some of the experiments not shown in Figure 1? Only 11 out of 17 experiments can be seen. Page 7 lines 1-4: This discussion is very unclear.

Page 7 line11: It is of course difficult to reproduce concentrations of VOC and NOx in an experiment, which is also clear from the present work. Therefore the statement of "similar" conditions is too strong. VOC concentration varies from 472 to 665 ppb, which also affects SOA formation to some degree (seen by plotting the data presented here). Page 7 line 18-19: This seems speculative. Page 7 line 22-23: This meaning is unclear.

Section 3.3. Page 7 line 30: According to your data in Table S1 the SOA yield without SO2 present was 2.7-3.4%, not 2.5-2.7% as stated here.

Page 8 line 10: The ratio was 4.4-7 according to Table S1, not 5 as stated here.

Figure 2. The SOA yield shows a trend, but only to some degree, since the values for low SO2 concentrations are somewhat scattered, while experiments at high concentrations have not been repeated. This means that conclusion about a trend in the data is based on very few data points at high SO2 concentrations. The discussion e.g. on page 8 lines 23-31 should be revised considerably with this in mind.

page 9 line 1-3: Which experiments were with NO and which with NO2?

Section 3.4 Please distinguish between sulfonate and organosulfate and make this more clear in the text. Figure 3: The line for ratio should be removed as it is based on very few data points. Figure 5: This figure is very confusing. Some data points are placed on top of each other. Information on the secondary axis (scale + label) is missing. Furthermore the uncertainty on the measurements should be presented. I suggest to make two figures instead of one.

Page 10 line 30- and Figure 6: Did the composition and response of organosulfates vary between samples?

In conclusion, some of the results are interesting, but the quality of the presentation and discussion is not adequate for publication of this work in ACP.

---

## Referee Comment (RC2) · Anonymous Referee #1 · 17 Mar 2017

This manuscript described photooxidation of cyclohexene with changing SO2 concentration and concluded that both SO2 competing OH and acid-catalyzed heterogeneous reactions are important to result in the observed SOA yield trend. Organosulfates are also observed in SOA composition. The main results are clearly stated, but the discussion could have been more comprehensive and in depth, regarding the complementary measurements. I think there are a few major issues that the authors need to address before the manuscript can be published.

Major comments:
1. As initial SO2 concentration increases, it becomes more competitive of OH radicals against cyclohexene, whose initial concentration remain stable. Note cyclohexene could also react with O3. From the Figure S1 in supplemental, it seems O3 concentration was already high at 20 min. If OH is mostly reacted with SO2, when cyclohexene remains, cyclohexene + O3 could become the dominant pathway for cyclohexene loss. At page 9, line 8-9, the authors argued that the ozonolysis rate is 6 orders of magnitude lower than OH oxidation. But the much higher O3 concentration than OH could offset this difference, especially in the case of the current study where OH was insufficient. Therefore, varying only initial SO2 could cause very different cyclohexene chemistry (OH oxidation vs. ozonolysis). I think this is important to discuss. The authors at least need to provide convincing data to show ozonolysis is not important in this study. A statement like "In addition to the kinetic limitation of the cyclohexene reaction with O3, the typical concentration of O3 in our chamber was 200 ppb and hence the importance of cyclohexene reaction with O3 was expected to be less significant than that of its reaction with OH under any relevant SO2 conditions" is too vague. The authors did not include cyclohexene concentration in the supplementary figure, which is an important indicator of the VOC chemistry. Also, it will be evident if the authors could show cyclohexene decay curves at different initial SO2 concentrations. It might be best if the authors could provide a figure (could include that in Figure 2) estimating how much cyclohexene reacts with OH vs. O3 under the studied SO2 concentrations.

2. Comparing Figure 3 and Figure 5, it is unclear to me what the amount of sulfate indicates in either case and they obviously represent different measurements. Is the amount of sulfate in Figure 3 only from IC measurement (inorganic $SO_4^{2-}$)? And the amount of sulfate in Figure 5 from IC measurement normalized to SOA mass? I think it is the main results of this manuscript and need to be clearly stated in the figure caption. Also I wonder how does the ratio of the FTIR band 1100 $cm^{-1}$ to IC $SO_4^{2-}$ as a function of initial SO2 look like? This ratio might tell how efficiently organosulfates are formed under changing SO2 concentrations.

3. As indicated by the title, I think more discussion regarding chemical composition is needed. From the results, only organosulfates are focused. From Figure S1, it looks NOx gets lost to organic nitrates. From Figure 4, the IR data suggest –ONO2 presents in SOA. I suggest the authors discuss more on organic nitrate in SOA. Only a paragraph at the very end seems insufficient. For example, does –ONO2 IR data correlate better with SOA yield? What N-containing chemical formulae present in the ESI-HR-MS data? Any suggested mechanisms?

4. Based on Page 8, Line 35, it seems both NO- and NO2-initiated experiments were conducted. But it is unclear according to Table S1. The authors used NOx in most experiment description. I think it is better to state clearly whether they used NO or NO2. The SOA yields might be similar, but chemistry and timescales of SOA formation might be different, as the authors already indicated.

Minor comments:

I have a big issue with the literature citing quality of this manuscript (and I do not know how to make suggestions because there are too many of those). Some examples: Page 1, Line 27. A few important review papers need to be cited in the first paragraph of introduction, such as the Halliquist et al. 2009 ACP, Kroll et al., 2008 AE. Page 8, Line 33. Many papers were published demonstrating acid-catalyzed heterogeneous reactions and enhanced SOA formation before and around 2010. The authors did not cite the most important studies.

Page 2, Line 1. Jaoui et al., 2012 citation was not in the reference list.

Page 4, Line 6. It should be specified, whether NO or NO2 was injected.

Page 4, Line 23. What were the TD temperature and time?

Page 5, Line 9. FTIR analysis uses 300L of air sample, >75% of total chamber volume. Discuss potential artifact.

Page 8. Line 8. It is problematic to say "NO3-initiated reaction was a poor source of SOA". Presto et al., 2005a and some later studies did find out that NO3 oxidation of alpha-pinene does not make a lot of SOA, but not necessarily for cyclohexene.

---

## Referee Comment (RC3) · Anonymous Referee #2 · 17 Mar 2017

This manuscript presents laboratory measurements on the photooxidation of cyclohexene, with a focus on the change of SOA yield and chemical composition as a function of SO2 concentrations. The authors concluded that competitive reaction of OH radicals with SO2 and VOCs was the main reason that dictates the cyclohexene SOA yields, and presented FTIR, IC, and ESI-HR-MS data to support the formation of organosulfates in this specific system. Overall, this study provides useful information relevant to a better understanding of cyclic alkene SOA formation. However, there are a few major concerns regarding the connections between the reported data and the speculated mechanisms that need to be addressed before publication can be considered. Also, more in depth discussions are needed to improve the current manuscript. Below I listed a few specific questions for the authors' clarification.

1) In the abstract line 14-17, these two sentences are very confusing and somewhat

contradictory with other statements in the manuscript. What is the real impact of acid catalyzed-mechanisms on cyclohexene SOA formation? 2) Did cyclohexene react completely in each experiment? The experimental profile presented in Figure S1 didn't include the traces of VOC precursor and main gas phase products. Since in section 2.2 the authors mentioned that these compounds were measured by TD-GC-MS measurements, these data should be included in discussion. 3) What is the connection between OH-limited scenario presented here (that leads to competitive reactions) and the real atmospheric environment? This is not clearly stated in the manuscript. 4) How does the chemical composition of SOA change in the absence versus in the presence of SO2? Without the initial input of SO2, the SOA yield was already substantial. It appears that with and without SO2 addition, SOA was formed through different pathways (homogeneous nucleation versus heterogeneous uptake/partitioning). This needs to be discussed in more detail. Also, the authors provided a full set of FT-IR spectra and sulfate concentrations. Are the corresponding ESI-MS data available? These will be useful to strengthen the discussion on organosulfate formation. 5) In the last paragraph of section 3.2, the authors stated that SOA yield was enhanced by acid-catalyzed heterogeneous reactions when SO2 concentrations are high. Is there direct evidence to support the proposed acid-catalyzed reactions? Was aerosol acidity measured or estimated? What is the potential role of particle sulfate contents for surface or bulk accommodation?

---

## Author Comment (AC1) · 24 May 2017

We have revised our manuscript according to the suggestions of the Referee's comments. For clarity, the Referee' comments are reproduced in blue, authors' responses are in black and changes in the manuscript are in red color text. Pages and lines of modified/inserted/deleted texts are relative to the previous version of the manuscript.

**Anonymous Referee #4**

This manuscript presents interesting new results on atmospheric reactions of photooxidation of cyclohexene in the presence of SO2. Unfortunately the quality of the presentation is not suitable for publication in ACP regarding scientific discussion and interpretation of results. The manuscript must be rewritten to discuss the results from a more objective, scientific point, which to a higher degree takes data uncertainty into account before reaching conclusions. Furthermore, there are many grammatical errors. In the following I have explained some of the major points.

In addition to the many comments responded below, the whole manuscript has been checked, several errors have been fixed, to improve the quality of the manuscript.

Introduction the introduction should include more references to relevant previous work. One example is in line 23 page 1, where only one study (on measurement methods for VOC from vehicles) is used as reference for the general statement about emission of alkenes from biogenic and anthropogenic sources.

We carefully reviewed the citations and several relevant previous works were added, some of which are given below:

Chin, J. Y., and Batterman, S. A.: VOC composition of current motor vehicle fuels and vapors, and collinearity analyses for receptor modeling, Chemosphere, 86, 951-958, doi: 10.1016/j.chemosphere.2011.11.017, 2012.

Hallquist, M., Wenger, J. C., Baltensperger, U., Rudich, Y., Simpson, D., Claeys, M., Dommen, J., Donahue, N. M., George, C., Goldstein, A. H., Hamilton, J. F., Herrmann, H., Hoffmann, T., Iinuma, Y., Jang, M., Jenkin, M. E., Jimenez, J. L., Kiendler-Scharr, A., Maenhaut, W., McFiggans, G., Mentel, T. F., Monod, A., Prevot, A. S. H., Seinfeld, J. H., Surratt, J. D., Szmigielski, R., and Wildt, J.: The formation, properties and impact of secondary organic aerosol: current and emerging issues, Atmos. Chem. Phys., 9, 5155-5236, doi: 10.5194/acpd-9-3555-2009 2009.

Hatch, L. E., Creamean, J. M., Ault, A. P., Surratt, J. D., Chan, M. N., Seinfeld, J. H., Edgerton, E. S., Su, Y. X., and Prather, K. A.: Measurements of isoprene-derived organosulfates in ambient aerosols by aerosol time-of-flight mass spectrometry-part 1: Single particle atmospheric observations in atlanta, Environ. Sci. Technol., 45, 5105-5111, doi: 10.1021/es103944a, 2011.

Kesselmeier, J., Kuhn, U., Rottenberger, S., Biesenthal, T., Wolf, A., Schebeske, G.,

Andreae, M. O., Ciccioli, P., Brancaleoni, E., Frattoni, M., Oliva, S. T., Botelho, M. L., Silva, C. M. A., and Tavares, T. M.: Concentrations and species composition of atmospheric volatile organic compounds (VOCs) as observed during the wet and dry season in Rondonia (Amazonia), J. Geophys. Res., 107, LBA 20-21–LBA 20-13, doi: 10.1029/2000jd000267, 2002.

Kroll, J. H., and Seinfeld, J. H.: Chemistry of secondary organic aerosol: Formation and evolution of low-volatility organics in the atmosphere, Atmos. Environ., 42, 3593-3624, doi: 10.1016/j.atmosenv.2008.01.003, 2008.

Paulson, S. E., Chung, M. Y., and Hasson, A. S.: OH radical formation from the gas-phase reaction of ozone with terminal alkenes and the relationship between structure and mechanism, J. Phys. Chem. A, 103, 8125-8138, doi: 10.1021/Jp991995e, 1999.

The reference Jimenez et al. (page 1 line 28) on reactions of polyfluorinated compounds is not relevant for a general statement on SOA formation in the atmosphere, and it should thus be removed.

This reference was deleted.

On page 2 (lines 27-30) it is stated that a substantial amount of organosulfates have been observed. Most measurements estimate up to 10% of aerosol mass, and typically much lower in most places, which in my opinion is not a "substantial amount".

The original meaning of "substantial amount" was wrong. Our aim was to point out that different organosulfates were observed. Hence, "substantial amount" has been changed to "different kinds".

Why so much focus on organosulfate formation from MBO, which is typically not found in high concentrations? Page 2 lines 11-21: This section has a good number of relevant references.

We agree that MBO is not among the major molecules responsible for organosulfates formation. From the literature, laboratory chamber studies showed that OH/NOx/$O_3$-initiated reactions of BVOCs, such as isoprene, α-pinene, β-pinene, and limonene with sulfates or sulfuric acid are the main formation processes for organosulfates formation. Although organosulfates formation from MBO photooxidation is not as important as from isoprene and pinene reactions, MBO emissions were found to be larger than isoprene emissions in some regions (Baker et al., 1999). Hence, organosulfates formation from MBO reactions would not be negligible in those conditions. To take this into account in the manuscript, the text at page 2 lines 24 to page 3 line 10 was modified as:

"Despite the existence of organosulfates in ambient aerosols was first observed in

2005 (Romero and Oehme, 2005), proper identification of these aerosols was made two years later. In a series of chamber experiments studies, it was shown that organosulfates present in ambient aerosols collected from various locations mostly originate from acid-catalyzed reactions of SOA formed from photooxidation of α-pinene and isoprene (Surratt et al., 2007). Recently, different kinds of organosulfates have been observed in SOA around the world, and organosulfates have been identified as a group of compounds that have an important contribution to the total amount of SOA in the atmosphere (Surratt et al., 2008; Froyd et al., 2010; Kristensen and Glasius, 2011; Tolocka and Turpin, 2012; Wang et al., 2015). Laboratory chamber studies showed that OH/NOx/O$_3$-initiated reactions of BVOCs, such as isoprene, α-pinene, β-pinene, and limonene with sulfates or sulfuric acid are the main processes for organosulfates formation (Surratt et al., 2007; Surratt et al., 2008; Hatch et al., 2011). Despite the well-recognized presence of organosulfates in SOA, their formation and transformation processes can be complex and varied, depending on the nature of the original organic compound involved. Extensive studies on their formation have been performed and several mechanisms based on a variety of reactions have been proposed. Using nuclear resonance techniques, isoprene-derived epoxides formed during isoprene photooxidation reactions were found to be important intermediates for organonitrates and organosulfates formation via potential SOA reactions (Darer et al., 2011; Hu et al., 2011). The authors further found that organonitrates could easily be transformed to organosulfates during hydrolysis in the presence of sulfate. Some studies also showed that 2-Methyl-3-buten-2-ol (MBO), due its larger emissions than isoprene in some regions (Baker et al., 1999), is an important precursor for organosulfates and SOA in the atmosphere, through its reactions with OH under NO and aerosol acidity conditions, and from acid-catalyzed reactive uptake of MBO-based epoxides formed during MBO photooxidation (Mael et al., 2015; Zhang et al., 2012; Zhang et al., 2014). Organosulfates formation was also found from oxidation of hydroxyhydroperoxides (Riva et al., 2016) and from heterogeneous reactions of SO$_2$ with selected long-chain alkenes and unsaturated fatty acids (Passananti et al., 2016)."

Experimental Page 4 line 8: The VOC/NOx ratio was not about 5, but varied from 4.4 to 7.

The ratio of VOCs/NOx has been fixed.
The sentence at page 4, line 8 "The mixed concentration ratios of VOCs/NOx were adjusted to be about 5" was changed to "The mixed concentration ratios of VOCs/NOx were in the range 4.4-6.9."

Section 2.3: Why were only aerosols in the range 108-650 nm collected? What was the limit of detection of the IC analysis?

The particles were collected by a Dekati low pressure impactor (DLPI, DeKati Ltd, Finland). The size range of particles collected by this device was 0.03-10μm, which was divided into thirteen segments. According to the results of particle size range measured by SMPS as shown in Figure R1 below, the mass of particles formed were in the range of 75-300 nm. These were in the 108-650 nm range, which corresponds to the third segment, within which most of the particles from our measurements could be collected. The particles in this range constitutes about 97% of the total mass, both measured (black bars) and simulated (red line).

[Figure]

Figure R1. Particle size distribution

For the IC analysis, the limit of detection was 0.2 mg mL$^{-1}$.
The following is added in the manuscript at page 5 line 13 to clarify:
"For the IC analysis, the limit of detection was 0.2 mg mL$^{-1}$."

Were samples for FTIR and SOA analysis collected right after each other?

No. While FTIR is very sensitive and does not require high concentrations of the samples, the SOA analysis needs higher concentrations. For SOA analysis, the concentrations were increased in order to have more SOA formation. Hence, the samples for SOA analysis were collected after the samples for FTIR analysis. However, both samples types were collected using the same method, and all experiments were conducted using the same procedure.

Results and discussion Section 3.1 should be moved to the experimental section.

We have moved the content of Section 3.1 to the experimental section.

Section 3.2 Page 6 line 22. The trend is not clear, especially regarding maximum particle number, which shows quite some scatter.

Page 6 line 26-32: The conclusions in this section are beyond what I see in the data, given the scatter.
Why are some of the experiments not shown in Figure 1? Only 11 out of 17 experiments can be seen.

Based on the available data and despite the scattered behavior of the maximum particle number concentration relative to the particle number concentration at maximum yield with increasing $SO_2$ concentration, we can find an acceptable correlation between the two types of number concentrations, as we mentioned at page 6 lines 21-22. The change in the trend of SOA number concentrations is quite evident from the particle number concentrations at the maximum SOA yield as can be observed in Figure R2 below.

[Figure]

Figure R2: Particle number concentrations of SOA in the photooxidation of the cyclohexene/NOx/SO$_2$ system with different initial SO$_2$ concentrations.

The scattered behavior of the maximum particle number concentrations was likely caused by measurement errors. These concentrations were obtained upon particles formation after the black light lamps were switched on. SOA number concentrations increased rapidly to reach the maximum, and subsequently, the particle number concentrations decreased, as shown in Figure S1. The particles were collected by SMPS at a sampling interval of 5 min. The decrease rate of the particle number concentration was very fast, about 25% after 5 min, which also corresponds to the time at which the maximum number concentration was obtained.
To avoid any misinterpretation of our data and our conclusions due to the data being scattered, Figure 1 of the main manuscript was replaced by Figure R2 in this response.

There are two main reasons why there are some maximum particle number concentrations were missing. Firstly, as shown in Figure S1, the maximum number concentration was obtained in the beginning of SOA formation, but the maximum mass concentration was obtained one hour later. For our experiments, we focus on the

ultimate yield of SOA and hence, the SMPS analysis was not performed immediately after the UV light was turned on.

Secondly, the collection volume by the impactor was slightly smaller than the volume of the chamber. In order to reduce gas consumption in the smog chamber, the particles were not sampled continuously by SMPS from the beginning of the experiment. Despite the missing data on the maximum particle number concentration the trend of SOA number concentration with initial $SO_2$ is obvious.

Page 7 lines 1-4: This discussion is very unclear.

We re-wrote this as:

"It is evident from Figure 1 that even small amounts of $SO_2$ affect the new particle formation substantially. This is in agreement with the finding that wood soot, a minor source of $SO_2$ (Reddy and Venkataraman, 2002), resulted in a measurable positive deviation to the VOCs/NOx photooxidation reaction system without background aerosol (Jang et al., 2002)."

Page 7 line11: It is of course difficult to reproduce concentrations of VOC and NOx in an experiment, which is also clear from the present work. Therefore the statement of "similar" conditions is too strong. VOC concentration varies from 472 to 665 ppb, which also affects SOA formation to some degree (seen by plotting the data presented here).

We agree that the statement with "similar" can be misleading. Because this statement does not actually give extra information on the stated finding, it was deleted, however, without changing the meaning of the main result.

Page 7 line 18-19: This seems speculative.

This sentence was deleted.

Page 7 line 22-23: This meaning is unclear.
This was re-written as:
"New particles were formed by vapor condensation onto existing aerosol particles."

Section 3.3. Page 7 line 30: According to your data in Table S1 the SOA yield without SO2 present was 2.7-3.4%, not 2.5-2.7% as stated here.

This was fixed.

Page 8 line 10: The ratio was 4.4-7 according to Table S1, not 5 as stated here.

This was fixed.

Figure 2. The SOA yield shows a trend, but only to some degree, since the values for low SO2 concentrations are somewhat scattered, while experiments at high concentrations have not been repeated. This means that conclusion about a trend in the data is based on very few data points at high SO2 concentrations. The discussion e.g. on page 8 lines 23-31 should be revised considerably with this in mind.

Due to the error associated with measuring $SO_2$ concentrations, many experiments were needed at low concentrations (below 40 ppb) for a better reproducibility of the experimental data since even 1 ppb error in $SO_2$ concentration can have significant effects. However, this error is less important at higher $SO_2$ concentrations and hence, few data points were used. The scattered behavior of the SOA yield at low $SO_2$ concentrations is primarily a consequence of the errors due to measuring the concentrations, and the trend of Figure 2 is not expected to change. Based on this, the discussion on page 8 lines 23-31 is modified as:

"SOA yields for the cyclohexene/NOx/$SO_2$ system were measured for initial $SO_2$ mixing ratios of 0-105 ppb. Due to the error associated with the $SO_2$ concentrations measurement, with stronger impact on low values than on higher values, several experiments were performed at $SO_2$ concentrations below 40 ppb. The experimental results showed a clear decrease at first and then an increase in the SOA yield with increasing $SO_2$ concentrations (Figure 2). When $SO_2$ concentrations increased from 0 to 40.8 ppb, there was a remarkable decrease in SOA yield, dropping by about half with the increase of $SO_2$ concentration. For $SO_2$ concentrations higher than 40.8 ppb, SOA yield increased with increasing $SO_2$ concentration. The highest SOA yield was obtained to be 3.5%, at 104.7 ppb $SO_2$ concentration. The lowest SOA yield of cyclohexene photooxidation was obtained at the initial $SO_2$ concentration of 40 ppb. Although the SOA yield increased gradually with the initial $SO_2$ concentration at concentrations higher than 40 ppb, a higher SOA yield than that in the absence of $SO_2$ could not be obtained when the initial $SO_2$ concentration was lower than 85 ppb."

Page 9 line 1-3: Which experiments were with NO and which with NO2?

A new plot for Figure 2 was made, in which experiments initiated with NO and $NO_2$ are clearly distinguished. Also, different NO- and $NO_2$-initiated experiments are marked in Table 1.

[Figure]

Figure 2: SOA yields of cyclohexene photooxidation in the presence of NOx at different initial $SO_2$ concentrations. Solid line is least-square fitting to the data. The error bars were determined on the basis of propagation of uncertainties arising in the $\Delta HC$ measurements, including GC calibration uncertainties propagation and the variance in the initial cyclohexene measurements.

Table 1 Experimental conditions for the photooxidation of cyclohexene/NOx/SO$_2$ system. All experiments were performed under dry conditions (relative humidity < 10%). $\Delta M_0$ is the produced organic aerosol mass concentration and Y is the SOA yield.

| Exp. | T (K) | SO$_2$ (ppb) | cyclohexene (ppb) | NOx (ppb) | cyclohexene/NOx | $\Delta M_0$ ($\mu$g m$^{-3}$) | Y (%) |
|---|---|---|---|---|---|---|---|
| 1 [b] | 308 | 0.0 | 596 | 122.0 | 4.9 | 57.0 | 2.66 |
| 2 [b] | 305 | 0.0 | 651 | 93.7 | 6.9 | 79.7 | 3.40 |
| 3 [b] | 309 | 2.4 | 553 | 95.7 | 5.8 | 62.6 | 3.15 |
| 4 [a] | 307 | 5.8 | 612 | 92.7 | 6.6 | 41.0 | 1.87 |
| 5 [a] | 309 | 9.3 | 599 | 93.5 | 6.4 | 48.1 | 2.23 |
| 6 [b] | 309 | 11.0 | 574 | 94.7 | 6.1 | 47.1 | 2.28 |
| 7 [b] | 309 | 23.0 | 514 | 90.5 | 5.7 | 42.6 | 2.30 |
| 8 [b] | 305 | 36.6 | 665 | 99.7 | 6.7 | 96.3 | 2.01 |
| 9 [b] | 308 | 40.8 | 472 | 91.4 | 5.2 | 22.6 | 1.33 |
| 10 [a] | 308 | 44.3 | 592 | 98.6 | 6.0 | 35.3 | 1.66 |
| 11 [b] | 305 | 55.0 | 497 | 113.0 | 4.4 | 77.3 | 2.16 |
| 12 [b] | 308 | 58.8 | 577 | 96.7 | 6.0 | 44.3 | 2.13 |
| 13 [a] | 309 | 60.8 | 626 | 102.0 | 6.1 | 43.9 | 1.95 |
| 14 [a] | 308 | 72.7 | 581 | 98.4 | 5.9 | 49.2 | 2.35 |
| 15 [b] | 306 | 90.0 | 543 | 99.6 | 5.4 | 102.0 | 2.62 |
| 16 [a] | 309 | 104.7 | 608 | 93.7 | 6.5 | 77.1 | 3.52 |
| 17 [bc] | 305 | 236.0 | 1048 | 198.0 | 5.3 | - | - |
| 18 [bc] | 306 | 93.7 | 1235 | 215 | 5.7 | - | - |

a: the experiment was initiated by NO.
b: the experiment was initiated by NO$_2$.
c: the formed particles were detected by ESI-HR-MS.

Section 3.4 Please distinguish between sulfonate and organosulfate and make this more clear in the text.

Sulfonate and organosulfates are all members of organosulfur compounds, with respective formulae R-SO$_3^-$ and R-O-SO$_3^-$, where R is an organic alkyl or aryl group. The two terms are distinguished in the text.

Figure 3: The line for ratio should be removed as it is based on very few data points. Figure 5: This figure is very confusing. Some data points are placed on top of each other. Information on the secondary axis (scale + label) is missing. Furthermore the uncertainty on the measurements should be presented. I suggest to make two figures instead of one.

The ratio line from Figure 3 was removed. The discussion at page 9 lines 30-32 was modified as:

"Figure 3 shows that the changes with initial $SO_2$ concentrations were not uniform between the $SO_4^{2-}$ concentration and $\Delta SO_2$, which indicates that besides $SO_4^{2-}$, other products were formed from $SO_2$."

In Figure 5, all the results were normalized to SOA mass. In order to compare the changes of the results with different initial $SO_2$ concentrations and the relationship between the different results at the same $SO_2$ concentration, all the results were set as 1 at the initial $SO_2$ concentration of 44 ppb. This led to some points being overlapped in Figure 5. For clarification, we used different colors, Figure 5 was remade as follows. We do not need the secondary axis here since the magnitudes of the band at 1100 $cm^{-1}$ and $SO_4^{2-}$ are similar. The results of $\Delta SO_2$ were measured before DPLI sampling, both the volume of the chamber and the sampling time being inconsistent with those of the measurement of the band at 1100 $cm^{-1}$ and $SO_4^{2-}$. Hence, in order to be more rigorous, we deleted the results on $\Delta SO_2$ in Figure 5.

[Figure]

Figure 5: The relative intensity of the FTIR band at 1100 $cm^{-1}$ (square) and the amount of $SO_4^{2-}$ (triangle) normalized to SOA mass. The 1100 $cm^{-1}$ band intensity and the amount of $SO_4^{2-}$ were divided by the formed SOA mass. Subsequently, the results of both FTIR band at 1100 $cm^{-1}$ and the amount of $SO_4^{2-}$ divided by SOA mass were set to 1 when the initial $SO_2$ concentration was 44.3 ppb.

To clarify these changes, the statement at page 10 lines 23-24 was deleted. The statement at page 10 lines 27-28 was changed to "Figure 5 show the inconsistency between the FTIR band at 1100 $cm^{-1}$ and the amount $SO_4^{2-}$ as the initial $SO_2$ concentration, which implies that the 1100 $cm^{-1}$ band originated not only from $SO_4^{2-}$, but also from other organosulfur compounds. These include organosulfates, which also have the S=O bond, and might contribute to the 1100 $cm^{-1}$ band in the FTIR

spectrum. The gap between the FTIR band at 1100 cm$^{-1}$ and $SO_4^{2-}$ can be attributed to the formation of organosulfates."

Page 10 line 30- and Figure 6: Did the composition and response of organosulfates vary between samples?

We investigated the composition and response of organosulfates between samples by performing another experiment with different initial $SO_2$ concentrations. We found that these did not vary much with the change in initial $SO_2$ concentrations, as can be seen in Figure S5 below.

[Figure]

Figure S5: Comparison of SOA ESI-HR-MS spectra with different initial $SO_2$ concentrations.

The following was added at page 11 line 18 for clarification.
"The ESI-HR-MS spectra of particles formed from two different initial $SO_2$ concentrations are shown in Figure S3. We found no obvious difference in the composition and response of organosulfates with different initial $SO_2$ concentrations. The relative intensity of m/z = 97, which corresponds to sulfate was set to 100% in both ESI-HR-MS spectra. The relative intensities of the organosulfates peaks in both spectra were almost unchanged regardless of the initial $SO_2$ concentration, indicating that the organosulfates yield was associated with sulfate content. Our result is consistent with the results of Minerath et al. and Hatch et al. who observed an increase in organosulfates yield with increasing sulfate concentration (Minerath and Elrod, 2009; Hatch et al., 2011). These observations demonstrate that particle sulfate content is likely a key parameter influencing organosulfates formation."

In conclusion, some of the results are interesting, but the quality of the presentation and discussion is not adequate for publication of this work in ACP.

**References**

[revised manuscript text omitted]

---

## Author Comment (AC2) · 24 May 2017

We have revised our manuscript according to the suggestions of the Referee's comments. For clarity, the Referee's comments are reproduced in blue, authors' responses are in black and changes in the manuscript are in red color text. Pages and lines of modified/inserted/deleted texts are relative to the previous version of the manuscript.

**Anonymous Referee #1**

This manuscript described photooxidation of cyclohexene with changing $SO_2$ concentration and concluded that both $SO_2$ competing OH and acid-catalyzed heterogeneous reactions are important to result in the observed SOA yield trend. Organosulfates are also observed in SOA composition. The main results are clearly stated, but the discussion could have been more comprehensive and in depth, regarding the complementary measurements. I think there are a few major issues that the authors need to address before the manuscript can be published.

Major comments:
1. As initial $SO_2$ concentration increases, it becomes more competitive of OH radicals against cyclohexene, whose initial concentration remain stable. Note cyclohexene could also react with $O_3$. From the Figure S1 in supplemental, it seems $O_3$ concentration was already high at 20 min. If OH is mostly reacted with $SO_2$, when cyclohexene remains, cyclohexene + $O_3$ could become the dominant pathway for cyclohexene loss. At page 9, line 8-9, the authors argued that the ozonolysis rate is 6 orders of magnitude lower than OH oxidation. But the much higher $O_3$ concentration than OH could offset this difference, especially in the case of the current study where OH was insufficient. Therefore, varying only initial $SO_2$ could cause very different cyclohexene chemistry (OH oxidation vs. ozonolysis). I think this is important to discuss. The authors at least need to provide convincing data to show ozonolysis is not important in this study. A statement like "In addition to the kinetic limitation of the cyclohexene reaction with $O_3$, the typical concentration of $O_3$ in our chamber was 200 ppb and hence the importance of cyclohexene reaction with $O_3$ was expected to be less significant than that of its reaction with OH under any relevant $SO_2$ conditions" is too vague.

We agree with the Referee that cyclohexene can be oxidized by both ozone and OH. Although SOA can be formed from ozonization, but $O_3$ cannot react with $SO_2$. Moreover, the ozone concentrations in every experiment with different initial $SO_2$ concentrations were almost unchanged. So, the effect of $SO_2$ on $O_3$ concentration was not obvious. In a similar study involving β-pinene, Sarrafzadeh et al. proved that the increasing SOA yield was likely due to the increase in OH concentration though the concentration of $O_3$ was not mentioned in their experiment (Sarrafzadeh et al., 2016). However, it is worth noting that $O_3$ was added in their experiment. It has also been proved that the OH yield from β-pinene reacting with $O_3$ was about 30% (Nguyen et al., 2009; Ma and Marston, 2008), and the OH yield from the reaction of $O_3$ with

cyclohexene was 60% (Presto and Donahue, 2004). In the study of Sarrafzadeh, when [VOCs]/[NOx] was about 5, the OH concentration in the chamber was about $3.7 \times 10^7$ $cm^{-3}$. This concentration was higher in our experiment. Furthermore, the ratio of [OH]/[$O_3$] was higher in our experiments than in Sarrafzadeh's experiments. All in all, these are in favor to our argument that the SOA formed from cyclohexene was likely due to the OH reaction, and the SOA yield was likely due to the changing of OH concentration in our study.

The authors did not include cyclohexene concentration in the supplementary figure, which is an important indicator of the VOC chemistry. Also, it will be evident if the authors could show cyclohexene decay curves at different initial $SO_2$ concentrations. It might be best if the authors could provide a figure (could include that in Figure 2) estimating how much cyclohexene reacts with OH vs. $O_3$ under the studied $SO_2$ concentrations.

We have patched new experiments. Figure S4 below shows the change of cyclohexene concentration with time, at different initial $SO_2$ concentrations.

[Figure]

Figure S4: Change of cyclohexene concentration with time at different initial $SO_2$ concentrations.

As shown in Figure S4, the reacted cyclohexene concentration at 0 ppb initial $SO_2$ concentration was slightly higher than that at 90 ppb. The consuming rate of cyclohexene was higher without $SO_2$ in the chamber, which means that if there was a competition reaction, its effect was not significant. Due to the sparse data of cyclohexene concentration in the experiment with 40 ppb initial $SO_2$, they could not be fitted. However, they fell between the fitted data at 0 and 90 ppb initial $SO_2$ concentration, being closer to the fit at 90 ppb. This further indicates that the presumed competition reaction was more obvious at low $SO_2$ concentrations than that at high $SO_2$ concentrations. The particle number concentration, which is related to the sulfate formed from $SO_2$ reaction with OH was also increased quickly at low $SO_2$ concentrations. This result explains why the SOA yield was decreased at low initial

SO$_2$ concentration as shown in Figure 3.

The following sentence was inserted at page 9 line 14.

"The change of cyclohexene concentration with time at different initial SO$_2$ concentrations is shown in Figure S4, wherefrom it can be seen that the reacted cyclohexene concentration at 0 ppb initial SO$_2$ concentration was slightly higher than that at 90 ppb. The consuming rate of cyclohexene was higher without SO$_2$ in the chamber, which means that if there was a competition reaction, its effect was very limited."

2. Comparing Figure 3 and Figure 5, it is unclear to me what the amount of sulfate indicates in either case and they obviously represent different measurements. Is the amount of sulfate in Figure 3 only from IC measurement (inorganic SO$_4^{2-}$)? And the amount of sulfate in Figure 5 from IC measurement normalized to SOA mass? I think it is the main results of this manuscript and need to be clearly stated in the figure caption. Also I wonder how does the ratio of the FTIR band 1100 cm$^{-1}$ to IC SO$_4^{2-}$ as a function of initial SO$_2$ look like? This ratio might tell how efficiently organosulfates are formed under changing SO$_2$ concentrations.

The amount of sulfate in Figure 3 is only from IC measurement (inorganic SO$_4^{2-}$), while the amount of sulfate from IC measurement in Figure 5 is normalized to the SOA mass, as stated at page 10, lines 18-21. For further clarifications, the caption of Figure 5 was modified to be:

"Figure 5: The relative intensity of the FTIR band at 1100 cm$^{-1}$ (square) and the amount of SO$_4^{2-}$ (triangle) normalized to SOA mass. The 1100 cm$^{-1}$ band intensity and the amount of SO$_4^{2-}$ were divided by the formed SOA mass. Subsequently, the results of both FTIR band at 1100 cm$^{-1}$ and the amount of SO$_4^{2-}$ divided by SOA mass were set to 1 when the initial SO$_2$ concentration was 44.3 ppb."

The intensity of FTIR band at 1100 cm$^{-1}$ can be used for quantitative analysis of the S=O bond in the particulate phase. The components having the S=O bond can be inorganic (sulfate) and organic (sulfate group in organosulfates). Although sulfate was quantified by IC, the relationship between SO$_4^{2-}$ concentration and FTIR band could not be determined. This means that the ratio function of the FTIR band at 1100 cm$^{-1}$ to IC SO$_4^{2-}$ here can only represent the relative variation of these two results, with no actual dimension.

As shown in Figure 5 in the main manuscript, when initial SO$_2$ concentrations are higher than 50 ppb, the opposite relationship between the 1100 cm$^{-1}$ band and the sulfate concentration normalized to SOA mass indicates that the band at 1100cm$^{-1}$ does not fully contribute to sulfate, but also to the sulfate group in organosulfates. The following sentence was inserted at page 10 line 27.

"Figures 5 shows the inconsistency between the FTIR band at 1100 cm$^{-1}$ and the amount of SO$_4^{2-}$ as the initial SO$_2$ concentration changes, which implies that the 1100 cm$^{-1}$ band originated not only from SO$_4^{2-}$, but also from other organosulfur

compounds. These include organosulfates, which also have the S=O bond, and might contribute to the 1100 cm$^{-1}$ band in the FTIR spectrum. The gap between the FTIR band at 1100 cm$^{-1}$ and SO$_4^{2-}$ can be attributed to the formation of organosulfates."

3. As indicated by the title, I think more discussion regarding chemical composition is needed. From the results, only organosulfates are focused. From Figure S1, it looks NOx gets lost to organic nitrates. From Figure 4, the IR data suggest –ONO2 presents in SOA. I suggest the authors discuss more on organic nitrate in SOA. Only a paragraph at the very end seems insufficient. For example, does –ONO2 IR data correlate better with SOA yield? What N-containing chemical formulae present in the ESI-HR-MS data? Any suggested mechanisms?

SOA formed by cyclohexene photooxidation was a mixture of different kinds of compounds, whose functional groups were assigned using FTIR as can be seen in page 10, lines 2-6. As shown in Figure 4, the bands at 1622 and 1230 cm$^{-1}$, assigned to –ONO$_2$, were the evidence for the existence of organonitrates. But, the intensity of –ONO$_2$ bands were very low, and this result was consisted with the ESI-HR-MS data that there was no evidence of the presence of N-containing compounds from the main peaks. The following was inserted at page 10 line 31 for clarification.

"The OH addition to the C=C bond produces an alkyl peroxyl (RO$_2$) radical that can react with NO to yield organonitrates (Perring et al., 2013). Although the formation of organonitrates was highly expected, there was no evidence of the presence of N-containing compounds from the main peaks of Figure 6, indicating that organonitrates would be formed at very low concentrations, if at all. A similar conclusion could be observed from Figure 4, when noticing that the -ONO$_2$ stretching peaks at 1622 and 1230 cm$^{-1}$ have very low intensities. The presumed low concentrations of organonitrates might be due to the low concentration of NO when SOA was formed. RO$_2$ radicals also react with NO$_2$ to form peroxy nitrates (RO$_2$NO$_2$) on time scales comparable to RONO$_2$ formation. However, RO$_2$NO$_2$ are thermally labile and rapidly dissociate at ambient temperatures (Perring et al., 2013)."

4. Based on Page 8, Line 35, it seems both NO- and NO2-initiated experiments were conducted. But it is unclear according to Table S1. The authors used NOx in most experiment description. I think it is better to state clearly whether they used NO or NO2. The SOA yields might be similar, but chemistry and timescales of SOA formation might be different, as the authors already indicated.

NO and NO$_2$ both initiated the experiments but, the photooxidation reaction could not happen in the case of NO until it was oxidized to NO$_2$, which means that both NO- and NO$_2$-initiated photooxidation reactions were actually triggered by NO$_2$. Hence, the chemistry of SOA formation from both processes is similar. It takes about 0.5 hour for NO to be fully converted to NO$_2$. Both O$_3$ and SOA formation occurred 0.5 later

in the NO-initiated photooxidation than in the $NO_2$-initiated photooxidation.

We have distinguished the NO- and $NO_2$-initiated experiments in Figure 2 and Table 1 and added related comments.

[Figure]

Figure 2: SOA yields of cyclohexene photooxidation in the presence of NOx at different initial $SO_2$ concentrations. The solid line is the least-square fitting to the data. The error bars were determined on the basis of propagation of uncertainties arising in the ΔHC measurements, including GC calibration uncertainties propagation and the variance in the initial cyclohexene measurements.

Table 1 Experimental conditions for the photooxidation of cyclohexene/NOx/SO$_2$ system. All experiments were performed under dry conditions (relative humidity < 10 %). $\Delta M_0$ is the produced organic aerosol mass concentration and Y is the SOA yield.

| Exp. | T (K) | SO$_2$ (ppb) | cyclohexene (ppb) | NOx (ppb) | cyclohexene/NOx | $\Delta M_0$ ($\mu g\ m^{-3}$) | Y (%) |
|------|-------|------|-------------|-----|-----------------|-----------------------|-------|
| 1 [b] | 308 | 0.0 | 596 | 122.0 | 4.9 | 57.0 | 2.66 |
| 2 [b] | 305 | 0.0 | 651 | 93.7 | 6.9 | 79.7 | 3.40 |
| 3 [b] | 309 | 2.4 | 553 | 95.7 | 5.8 | 62.6 | 3.15 |
| 4 [a] | 307 | 5.8 | 612 | 92.7 | 6.6 | 41.0 | 1.87 |
| 5 [a] | 309 | 9.3 | 599 | 93.5 | 6.4 | 48.1 | 2.23 |
| 6 [b] | 309 | 11.0 | 574 | 94.7 | 6.1 | 47.1 | 2.28 |
| 7 [b] | 309 | 23.0 | 514 | 90.5 | 5.7 | 42.6 | 2.30 |
| 8 [b] | 305 | 36.6 | 665 | 99.7 | 6.7 | 96.3 | 2.01 |
| 9 [b] | 308 | 40.8 | 472 | 91.4 | 5.2 | 22.6 | 1.33 |
| 10 [a] | 308 | 44.3 | 592 | 98.6 | 6.0 | 35.3 | 1.66 |
| 11 [b] | 305 | 55.0 | 497 | 113.0 | 4.4 | 77.3 | 2.16 |
| 12 [b] | 308 | 58.8 | 577 | 96.7 | 6.0 | 44.3 | 2.13 |
| 13 [a] | 309 | 60.8 | 626 | 102.0 | 6.1 | 43.9 | 1.95 |
| 14 [a] | 308 | 72.7 | 581 | 98.4 | 5.9 | 49.2 | 2.35 |
| 15 [b] | 306 | 90.0 | 543 | 99.6 | 5.4 | 102.0 | 2.62 |
| 16 [a] | 309 | 104.7 | 608 | 93.7 | 6.5 | 77.1 | 3.52 |
| 17 [bc] | 305 | 236.0 | 1048 | 198.0 | 5.3 | - | - |
| 18 [bc] | 306 | 93.7 | 1235 | 215 | 5.7 | - | - |

a: the experiment was initiated by NO.

b: the experiment was initiated by NO$_2$.

c: the formed particles were detected by ESI-HR-MS.

Moreover, Table S1 was moved to the main manuscript as Table 1. Further, the following was inserted at page 9, line 1:

"Although the photooxidation reaction could not happen in the case of NO until it was oxidized to NO$_2$, which means that both NO- and NO$_2$-initiated photooxidation reactions were actually triggered by NO$_2$, the chemistry of SOA formation from both processes is similar."

Minor comments:

I have a big issue with the literature citing quality of this manuscript (and I do not know how to make suggestions because there are too many of those). Some examples: Page 1, Line 27. A few important review papers need to be cited in the first paragraph of introduction, such as the Halliquist et al. 2009 ACP, Kroll et al., 2008 AE. Page 8, Line 33. Many papers were published demonstrating acid-catalyzed heterogeneous reactions and enhanced SOA formation before and around 2010. The authors did not

cite the most important studies.

This was fixed.

Page 2, Line 1. Jaoui et al., 2012 citation was not in the reference list.

Jaoui et al., 2012 citation was in the reference list at page 13, line 32.

Page 4, Line 6. It should be specified, whether NO or NO2 was injected.

We have distinguished the NO- and $NO_2$-initiated experiments in Figure 2 and Table 1. This is further detailed in our response to comment number 4.

Page 4, Line 23. What were the TD temperature and time?

The TD temperature was 280 $^{\circ}$C, and the sampling time was 3 min. The following sentence was inserted at page 4 line 25 to clarify.
"The TD temperature was 280 $^{\circ}$C, and the sampling time was 3 min."

Page 5, Line 9. FTIR analysis uses 300L of air sample, >75% of total chamber volume. Discuss potential artifact.

When the air sample was collected for FTIR analysis, and the volume of chamber was reduced, specific surface area of chamber decreased, and consequently, the wall loss of particles increased. However, SOA sampling was started after the maximum mass concentration was observed, and the change of chamber volume had no effect on the result of SOA yield. Although wall loss was increased, the collected SOA was still consistent with SOA yield between different experiments because of the same volume of sampling air.

Page 8. Line 8. It is problematic to say "$NO_3$-initiated reaction was a poor source of SOA". Presto et al., 2005a and some later studies did find out that NO3 oxidation of alpha-pinene does not make a lot of SOA, but not necessarily for cyclohexene.

The sentence "$NO_3$-initialed reaction was not a poor source of SOA for all kinds of VOCs" was deleted in order to cancel the contradiction.

**References**

Ma, Y., and Marston, G.: Multifunctional acid formation from the gas-phase ozonolysis of beta-pinene, Phys. Chem. Chem. Phys., 10, 6115-6126, doi: 10.1039/b807863g, 2008.
Nguyen, T. L., Peeters, J., and Vereecken, L.: Theoretical study of the gas-phase ozonolysis of

beta-pinene ($C_{10}H_{16}$), Phys. Chem. Chem. Phys., 11, 5643-5656, 2009.

Perring, A. E., Pusede, S. E., and Cohen, R. C.: An observational perspective on the atmospheric impacts of alkyl and multifunctional nitrates on ozone and secondary organic aerosol, Chem. Rev., 113, 5848-5870, doi: 10.1021/cr300520x, 2013.

Presto, A. A., and Donahue, N. M.: Ozonolysis fragment quenching by nitrate formation: the pressure dependence of prompt OH radical formation, J. Phys. Chem. A, 108, 9096-9104, 2004.

Sarrafzadeh, M., Wildt, J., Pullinen, I., Springer, M., Kleist, E., Tillmann, R., Schmitt, S. H., Wu, C., Mentel, T. F., and Zhao, D.: Impact of NOx and OH on secondary organic aerosol formation from β-pinene photooxidation, Atmos. Chem. Phys., 16, 11237-11248, doi: 10.5194/acp-16-11237-2016, 2016.

---

## Author Comment (AC3) · 24 May 2017

We have revised our manuscript according to the suggestions of the Referee's comments. For clarity, the Referee's comments are reproduced in blue, authors' responses are in black and changes in the manuscript are in red color text. Pages and lines of modified/inserted/deleted texts are relative to the previous version of the manuscript.

**Anonymous Referee #2**

This manuscript presents laboratory measurements on the photooxidation of cyclohexene, with a focus on the change of SOA yield and chemical composition as a function of SO2 concentrations. The authors concluded that competitive reaction of OH radicals with SO2 and VOCs was the main reason that dictates the cyclohexene SOA yields, and presented FTIR, IC, and ESI-HR-MS data to support the formation of organosulfates in this specific system. Overall, this study provides useful information relevant to a better understanding of cyclic alkene SOA formation. However, there are a few major concerns regarding the connections between the reported data and the speculated mechanisms that need to be addressed before publication can be considered. Also, more in depth discussions are needed to improve the current manuscript. Below I listed a few specific questions for the authors' clarification.

1) In the abstract line 14-17, these two sentences are very confusing and somewhat contradictory with other statements in the manuscript. What is the real impact of acid catalyzed-mechanisms on cyclohexene SOA formation?

Acid catalyzed reactions have been extensively proved to promote SOA formation (Jang et al., 2002; Jang and Kamens, 2002). At low $SO_2$ concentrations, the decreasing SOA yield might be due to the promoting effect of acid-catalyzed reactions on SOA formation. This effect was less important than the inhibiting effect of decreasing OH concentration, which was caused by the competition reaction of OH reactions with $SO_2$ and cyclohexene.

Specifically, heterogeneous uptake is responsible for aerosol mass increase in the presence of acid seed aerosol. Sulfate oxidant from $SO_2$ was the source of seed aerosol in our experiments. The oxygenated products of cyclohexene photooxidation including carbonyl and aldehyde group, which are able to react heterogeneously (Aschmann et al., 2012), are rapidly converted to low volatility products assigned to the particulate phase and increase the production of SOA (Cao and Jang, 2007). This additional accommodation of gas phase aldehydes to the particle phase progresses until no further heterogeneous reactions take place. Figure 5 shows that the sulfate amounts in unit mass of aerosols gradually decrease at high $SO_2$ initial concentration, which means that SOA formation can be promoted in acidic conditions.

To clarify the real effect of acid-catalyzed mechanisms in cyclohexene SOA formation, we modified the text in the Abstract lines 13-16 as:

"The decreasing SOA yield might be due to the fact that the promoting effect of acid-catalyzed reactions on SOA formation was less important than the inhibiting

effect of decreasing OH concentration at low initial SO$_2$ concentrations, caused by the competition reactions of OH with SO$_2$ and cyclohexene."

2) Did cyclohexene react completely in each experiment? The experimental profile presented in Figure S1 didn't include the traces of VOC precursor and main gas phase products. Since in section 2.2 the authors mentioned that these compounds were measured by TD-GC-MS measurements, these data should be included in discussion.

Cyclohexene reacted completely in each of our experiments. Its concentration was measured at the beginning and at the end of each experiment. We have patched experiments for the changing trend of cyclohexene during reaction. Figure S4 below shows the change of cyclohexene concentration with time, at different initial SO$_2$ concentrations.

[Figure]

Figure S4: Change of cyclohexene concentration with time at different initial SO$_2$ concentrations.

As shown in Figure S4, the reacted cyclohexene concentration at 0 ppb initial SO$_2$ concentration was slightly higher than that at 90 ppb. The consuming rate of cyclohexene was higher without SO$_2$ in the chamber, which means that if there was a competition reaction, its effect was not significant. Due to the sparse data of cyclohexene concentration in the experiment with 40 ppb initial SO$_2$, they could not be fitted. However, they fell between the fitted data at 0 and 90 ppb initial SO$_2$ concentration, being closer to the fit at 90 ppb. This further indicates that the presumed competition reaction was more obvious at low SO$_2$ concentrations than that at high SO$_2$ concentrations. The particle number concentration, which is related to the sulfate formed from SO$_2$ reaction with OH was also increased quickly at low SO$_2$ concentrations. This result explains why the SOA yield was decreased at low initial SO$_2$ concentration as shown in Figure 3.

The following sentence was inserted at page 9 line 14.

"The change of cyclohexene concentration with time at different initial SO$_2$ concentrations is shown in Figure S4, wherefrom it can be seen that the reacted

cyclohexene concentration at 0 ppb initial $SO_2$ concentration was slightly higher than that at 90 ppb. The consuming rate of cyclohexene was higher without $SO_2$ in the chamber, which means that if there was a competition reaction, its effect was very limited."

Only the cyclohexene concentration could be monitored with the GC-MS while other gaseous products could not, probably due to their low concentration not allowing their detection by the experimental device, and the selective adsorption of the Tenax tube. For more clarifications on this, the sentence at page4, line 21 was modified as:
"The concentrations of cyclohexene were analyzed by thermal desorption-gas chromatography-mass spectrometry (TD-GC-MS)."

We also did not find the organosulfates from the GC-MS spectra. Traditional analytical methods, such as GC-MS with prior derivatization, may not be well suited to identify organosulfates. It is likely that single derivatization protocols, such as trimethylsilylation, GC injection and column temperature could cause the degradation or misinterpretation of such species (Murray and Baillie, 1979). On the other hand, ESI-MS has been shown as an effective method for the detection of organosulfates species (Boss et al., 1999; Metzger et al., 1995). In this regard, only ESI-HR-MS data, as presented in Figure 6, were used for particle chemical composition discussion in this manuscript.

3) What is the connection between OH-limited scenario presented here (that leads to competitive reactions) and the real atmospheric environment? This is not clearly stated in the manuscript.

The competitive reaction of OH with $SO_2$ and cyclohexene can be important in environment enriched with $O_3$. For example, when OH reacts with $SO_2$ while cyclohexene and $O_3$ are present, cyclohexene $+ O_3$ reaction will become the dominant pathway for cyclohexene loss. Since SOA was mainly formed from the reaction of cyclohexene with OH and $SO_2$, the OH competition reaction would then lead to less SOA forming. In real atmospheric situations where $O_3$ is found in much higher proportion than OH, more cyclohexene will react with $O_3$ to form Criegee intermediates, which are good $SO_2$ oxidizers. Hence, even less SOA would form. However, this was not the case in our chamber. To clarify the atmospheric implications of competitive reactions, we inserted the following at page 9 line 14:
"Moreover, in real atmospheric situations where $O_3$ is found in much higher proportion than OH, cyclohexene would mainly react with $O_3$ to produce Criegee intermediates, which are good $SO_2$ oxidizers, and significantly less SOA than in the chamber will be formed."

4) How does the chemical composition of SOA change in the absence versus in the

presence of $SO_2$? Without the initial input of $SO_2$, the SOA yield was already substantial. It appears that with and without $SO_2$ addition, SOA was formed through different pathways (homogeneous nucleation versus heterogeneous uptake/partitioning). This needs to be discussed in more detail. Also, the authors provided a full set of FT-IR spectra and sulfate concentrations. Are the corresponding ESI-MS data available? These will be useful to strengthen the discussion on organosulfates formation.

Different particles were formed from cyclohexene photooxidation, and the particle chemical composition was very complex. Because of this complexity, the chemical composition could not be completely determined, making the understanding of the photooxidation mechanism incomplete. The overall chemical composition of particles was analyzed by FTIR, and it was found that the relative intensity of each characteristic peak did not show obvious change under different initial $SO_2$ concentrations. This means that in addition to organic sulfate formation, the remainder of the chemical composition is almost the same regardless of the initial $SO_2$ concentration.

For clarification, the sentence at page 10, lines 14-15 was modified as:

"However, the band of sulfate at 1100 $cm^{-1}$ in IR spectra increases with the rise of initial $SO_2$ concentration rather than the SOA yield, which suggests the formation of sulfonic acid group and sulfate product from $SO_2$ photooxidation since, only the relative difference in the intensities of FTIR peaks were studied here."

Generated particles were collected on ZnSe, and then detected by FTIR. During ESI-HR-MS detection, particles were collected on the aluminum foil using the same method as FTIR analysis and then extracted with 1 mL of acetonitrile. Considering the volume of the chamber and the volume of particles collected, corresponding ESI-MS data and FT-IR spectra for each experiment were not available. In order to afford more information about organosulfates composition, we performed one more experiment with different initial $SO_2$ to study the composition of organosulfates under different $SO_2$ concentrations, and the HR-MS result is shown in Figure S3 to appear in the Supplementary material. It is seen from this figure that the composition and response of organosulfates vary weakly with change in initial $SO_2$ concentrations.

[Figure]

Figure S5: Comparison of SOA ESI-HR-MS spectra with different initial $SO_2$ concentrations.

The following text was added at page 11 line 18 to strengthen the discussion on the composition and response of organosulfates with change in initial $SO_2$ concentrations: "The ESI-HR-MS spectra of particles formed from two different initial $SO_2$ concentrations are shown in Figure S5. We found no obvious difference in the composition and response of organosulfates with different initial $SO_2$ concentrations. The relative intensity of m/z = 97, which corresponds to sulfate was set to 100% in both ESI-HR-MS spectra. The relative intensities of the organosulfates peaks in both spectra were almost unchanged regardless of the initial $SO_2$ concentration, indicating that the organosulfates yield was associated with sulfate content. Our result is consistent with the results of Minerath et al. and Hatch et al. who observed an increase in organosulfates yield with increasing sulfate concentration (Minerath and Elrod, 2009; Hatch et al., 2011). These observations demonstrate that particle sulfate content is likely a key parameter influencing organosulfates formation."

5) In the last paragraph of section 3.2, the authors stated that SOA yield was enhanced by acid-catalyzed heterogeneous reactions when SO2 concentrations are high. Is there direct evidence to support the proposed acid-catalyzed reactions? Was aerosol acidity measured or estimated? What is the potential role of particle sulfate contents for surface or bulk accommodation?

We did not measure the aerosol acidity due to the limitations of our experimental equipment. Although, aerosol acidity correlated well with sulfate and could be estimated using the same way as Zhou et al. (Zhou et al., 2012), Zhou et al. pointed that when ambient RH is lower than the deliquescence point (DRH), the particle is considered to exist as a pure solid phase. Czoschke et al. pointed that a catalytic process takes place by a small amount of acid catalyst (5 µg m$^{-3}$) in dry conditions

(RH<10%) (Czoschke et al., 2003). As Figure 3 shows, the concentration of sulfuric acid in the chamber was greater than 5 μg m$^{-3}$ when the initial SO$_2$ concentration was greater than 40 ppb. This indicates that acid-catalyzed reactions took place in our experiment.

Sulfate oxidant from SO$_2$ as the source of aerosol seed in our experiments, contributes to the increase of the aerosol mass through heterogeneous uptake. The following was inserted at page 9 line 29 for clarifications:

"It was demonstrated that acid-catalyzed processes could take place when there is a small amount of acid catalyst (5 μg m$^{-3}$) (Czoschke et al., 2003). In our chamber, the concentration of sulfuric acid was greater than 5 μg m$^{-3}$ when the initial SO$_2$ concentration was greater than 40 ppb. This indicates that acid-catalyzed reactions were evident in our experiment."

**References**

Boss, B., Richling, E., Herderich, R., and Schreier, P.: HPLC-ESI-MS/MS analysis of sulfated flavor compounds in plants, Phytochemistry, 50, 219-225, doi: Doi 10.1016/S0031-9422(98)00526-3, 1999.

Czoschke, N. M., Jang, M., and Kamens, R. M.: Effect of acidic seed on biogenic secondary organic aerosol growth, Atmos. Environ., 37, 4287-4299, doi: 10.1016/S1352-2310(03)00511-9, 2003.

Hatch, L. E., Creamean, J. M., Ault, A. P., Surratt, J. D., Chan, M. N., Seinfeld, J. H., Edgerton, E. S., Su, Y., and Prather, K. A.: Measurements of isoprene-derived organosulfates in ambient aerosols by aerosol time-of-flight mass spectrometry - part 1: single particle atmospheric observations in Atlanta, Environ. Sci. Technol., 45, 5105-5111, doi: 10.1021/es103944a, 2011.

Jang, M., Czoschke, N. M., Lee, S., and Kamens, R. M.: Heterogeneous atmospheric aerosol production by acid-catalyzed particle-phase reactions, Science, 298, 814-817, doi: 10.1126/science.1075798, 2002.

Jang, M., and Kamens, R. M.: Atmospheric secondary aerosol formation by heterogeneous reactions of aldehydes in the presence of a sulfuric acid aerosol catalyst, 35, 4758-4766, 2002.

Metzger, K., Rehberger, P. A., Erben, G., and Lehmann, W. D.: Identification And Quantification Of Lipid Sulfate Esters by Electrospray-Ionization Ms/Ms Techniques - Cholesterol Sulfate, Analytical Chemistry, 67, 4178-4183, doi: Doi 10.1021/Ac00118a022, 1995.

Minerath, E. C., and Elrod, M. J.: Assessing the potential for diol and hydroxy sulfate ester formation from the reaction of epoxides in tropospheric aerosols, Environ. Sci. Technol., 43, 1386-1392, 2009.

Murray, S., and Baillie, T. A.: Direct Derivatization Of Sulfate Esters for Analysis by Gas-Chromatography Mass-Spectrometry, Biomed Mass Spectrom, 6, 82-89, doi: DOI 10.1002/bms.1200060209, 1979.

Zhou, Y., Xue, L. K., Wang, T., Gao, X. M., Wang, Z., Wang, X. F., Zhang, J. M., Zhang, Q. Z., and Wang, W. X.: Characterization of aerosol acidity at a high mountain site in central eastern China, Atmos. Environ., 51, 11-20, doi: 10.1016/j.atmosenv.2012.01.061, 2012.

---

## Author Response (AR2)

We have revised our manuscript according to the suggestions of the Referee's comments. For clarity, the Referee's comments are reproduced in blue, authors' responses are in black and changes in the manuscript are in red color text. Pages and lines of modified/inserted/deleted texts are relative to the previous version of the manuscript.

**Report #1**

1. The authors responded that heterogeneous uptake of gas phase aldehydes to the particle phase is responsible for aerosol mass increase in the presence of acid seed aerosol in this study through acid-catalyzed reactions. I am still confused about this part. Is there any direct evidence to support that such reactions do occur in this study? If so, what are the resultant SOA products in the particles? Are they measureable with FTIR or ESI-HRMS? I don't see any supporting data for this statement. The decrease of inorganic sulfate concentration under conditions of high SO2 concentration is understandable, since formation of organosulfate has been observed, which means that part of the inorganic sulfate contents has been incorporated into the organic molecules.

When studying the effect on acidic seed on the growth of isoprene- and α-pinene-based SOA, it was shown that FTIR peaks at 1180, 1050 and 879 $cm^{-1}$ are indicators of acid-catalyzed heterogeneous reactions since these peaks could not otherwise be observed in non-acidic conditions (Czoschke et al., 2003 and references therein). In our experiment, although the FTIR peak below 950 $cm^{-1}$ could not be obtained, two bands similar to 1180 and 1050 $cm^{-1}$ were observed, which supports the presence of acid-catalyzed reactions in our experiment. Their intensity was very weak when initial $SO_2$ concentrations were lower than 44 ppb. Although we were not able to identify the SOA products of acid-catalyzed reactions in our experiments, the resulting bands were detected with FTIR.
We added the following at page 9, line 2.
When studying the effect on acidic seed on the growth of isoprene- and α-pinene-based SOA, it was shown that FTIR peaks at 1180, 1050 $cm^{-1}$ are indicators of acid-catalyzed heterogeneous reactions since these peaks could not otherwise be observed in non-acidic conditions (Czoschke et al., 2003 and references therein). In our experiment, two similar bands located at 1195 and 1040 $cm^{-1}$, were observed (see Figure S4), which supports the presence of acid-catalyzed reactions in our experiment. These observed peaks are prominent in IR spectra from SOA formation in an acidic particle environment (Jang et al., 2002; Czoschke et al., 2003). In our study, the intensities of these peaks were very weak when initial $SO_2$ concentrations were lower than 44 ppb, indicating that acid-catalyzed reactions are not facilitated at these conditions.

[Figure]

Figure S4: The change of the two peaks at 1195 and 1040 cm$^{-1}$ at different initial SO$_2$ concentrations.

2. It is still unclear to me about chemical composition of SOA change in the absence versus in the presence of SO2. In Figure S5, the authors provided ESI-HR-MS spectra with different initial SO2 concentrations (236 ppb versus 93.7 ppb), and stated that no obvious composition change under these conditions. However, these are both under high SO2 conditions. Without the initial input of SO2 (i.e. initial SO2=0 ppb), the SOA yield was already significant, but acid-catalyzed reactions and organosulfate formation would not be expected under such conditions. This points to different SOA formation mechanisms that determine the SOA composition and eventually the SOA yield under high SO2 and zero SO2 conditions. In this context, it would be more meaningful to present ESI-HR-MS data from 0 ppb, 40 ppb, and 100 ppb to discuss how composition change may affect the SOA yield.

More HR-MS analyses were performed, with 0 ppb and 236 ppb initial SO$_2$ concentrations (see Figure S6). As can be seen from this Figure, the SOA composition changes with initial SO$_2$ concentrations.

[Figure]

Figure S6: Comparison of SOA ESI-HR-MS spectra with different initial $SO_2$ concentrations.

We added the following at page 12 line 13.

Comparing HR-MS data when initial concentrations were 0 ppb and 236 ppb reveals that the bands representing organosulfates do not appear at 0 ppb of $SO_2$. Peaks at m/z larger than 150 were undetectable at initial $SO_2$ concentration of 0 ppb, while products without sulfur peaked at both concentrations, with the only difference being their intensities. This implies that the process of SOA formation strongly depends on initial $SO_2$ concentrations.

3. The enhancement of organosulfate yield with increasing initial SO2 concentrations should be supported with quantitative or semi-quantitative ESI-HR-MS data with authentic or surrogate standards. It is unclear that how the relative intensity of m/z = 97 and the peak of organosulfate species could serve as an indicator for increasing organosulfate formation.

Organosulfates could be identified by ESI-HR-MS, but their concentrations should be measured using liquid chromatography (LC). The HR-MS results were obtained through a direct injection as mentioned in page 6 line 4. However, since LC was not combined to HR-MS, quantitative analysis of the SOA chemical composition was not made and surrogate standards were not used. All MS signals could be obtained at the same time, as shown in the figure below.

[Figure]

Figure S7: Raw HR-MS data as shown in the Xcalibur 2.2 software. The total ion chromatographic prints of ESI-HR-MS (top) and the mass spectra (bottom).

The HR-MS signals of organosulfates were related to those of sulfates as shown in Figure S6 and S7, which means the concentrations ratios of organosulfates and sulfates were consistent. Here, the amount of sulfates formed increased with increasing initial $SO_2$ concentration. Since the relative intensities of sulfates and organosulfates are similar, it follows that organosulfates concentrations increase when the $SO_2$ concentration increases.

We added the sentence below at page 12 line 13 to clarify.

Given the consistency between sulfates and organosulfates concentrations as shown in Figure S6 and S7, it is most likely that as the amount of sulfate increase with increasing initial $SO_2$ concentration, the concentration of organosulfates will also increase.

4. The authors concluded that the competitive reactions of OH with SO2 and cyclohexene is the main reason for the change of SOA yield under 40 ppb of SO2 initial concentration. However, this statement is not supported by any kinetic data or estimates based on published rate constants and measured reactant/product concentrations. Same for the reactions for the reaction of O3+cyclohexene, what are the lifetimes of cyclohexene against OH versus O3 oxidation? The authors should at least provide some estimates before reaching the conclusion of competitive reactions, as well as completely ignoring the potential contribution of SOA generated from O3+cyclohexene in this study.

Although about 40% of total cyclohexene was estimated to react with $O_3$, the proportion of SOA formed from this reaction was not clear. From Sarrafzadeh's study,

the SOA yield was considered as a function of OH concentration (Sarrafzadeh et al., 2016). Compared to Sarrafzadeh's study, a lower $O_3$ concentration was used in our study, and hence the contribution of $O_3$ reaction to SOA might be less important. In addition, from available rate constants, the lifetime of cyclohexene based on the reaction with $O_3$ (5.5 h) is more than twice higher than the lifetime based on the reaction with OH (2.5 h). This further indicates that the reaction with $O_3$ might be less important.

For clarification, we added the following at page 9 line 8.

The rate constants of $O_3$ + cyclohexene and OH + cyclohexene reactions were determined to be $7.44 \times 10^{-17}$ and $6.09 \times 10^{-11}$ $cm^3$ molecule$^{-1}$ s$^{-1}$, respectively, corresponding to 5.5 h and 2.5 h lifetimes for cyclohexene (Treacy et al., 1997; Rogers, 1989). Hence, it is likely that the cyclohexene reaction with $O_3$ would be less important than the reaction with OH in our study.

The text below was added at page 9 line 23.

The rate constant for the OH + $SO_2$ reaction was estimated to be $2.01 \times 10^{-12}$ $cm^3$ molecule$^{-1}$ s$^{-1}$, corresponding to a $SO_2$ lifetime of 69 h (Atkinson et al., 1997). This reaction is much slower than the cyclohexene + OH reaction, suggesting that OH + $SO_2$ reaction has very little impact on the OH concentration in the system. In our experiment, the decrease in the SOA yield with $SO_2$ addition might then not be attributed to its reaction with OH. It is also possible that the $SO_2$ addition could change the chemistry of the photooxidation process and suppress the oxygenation of products (Friedman et al., 2016; Liu et al., 2015). Comparing the HR-MS results at different initial $SO_2$ concentrations, the proportion of low molecular weight components increases with increasing $SO_2$ concentration. Molecular weights have negative correlation with volatility, which could also make the SOA yield to decrease.


We have revised our manuscript according to the suggestions of the Referee's comments. For clarity, the Referee's comments are reproduced in blue, authors' responses are in black and changes in the manuscript are in red color text. Pages and lines of modified/inserted/deleted texts are relative to the previous version of the manuscript.

**Report #2**

The authors have tried to improve the quality of the manuscript, but unfortunately some of the changes raise additional questions. Generally the quality of citations and figures have improved, but unfortunately new problems with correct citation of literature has developed.
I will try to limit my comments to the responses to my previous questions.

Page 5 line 25. The authors have now stated the detection limit:
"For the IC analysis, the limit of detection was 0.2 mg mL-1. "
I assume that mg should be corrected to microgram. Is this assumption correct?
The corresponding air concentration would be:
0.2 µg/mL * 7 mL extraction volume = 1.4 µg.
The sampling air volume was 0.3 m3, and the detection limit was thus 1.4 µg/0.3 m3 = 4.7 µg/m3.
So the detection limit is 4.7 microgram/m3, which is around the concentration for many measurements shown in Figure 3.
The authors must explain and discuss the uncertainty of the IC analyses. If my calculation above is correct, the IC analyses are highly uncertain.

We made a mistake here. We have checked and updated the detection limit (0.005 µg mL$^{-1}$).

Page 6 line 31: "This is in agreement with the finding that wood soot, a minor source of SO2 (Reddy and Venkataraman, 2002), resulted in a measurable positive deviation to the VOCs/NOx photooxidation reaction system without background aerosol (Jang et al., 2002)."
The authors have tried to make the sentence clearer, but not yet succeeded. First of all it is not clear how wood soot can be a source of SO2 gas, but the whole sentence makes more sense when it is compared with the original sentences from the citation.
Jang et al., 2002 wrote: "Compared to fossil fuels, biomass combustion is a minor source of SO2 (28). Yet wood soot also resulted in a measurable positive deviation from theoretical partitioning compared to the photooxidation reaction system of a-pinene and NOx without background aerosol."
Here it is clear that it is not wood soot, but biomass combustion, which is a minor source of SO2 in the environment in general.
In the present context it would be much more relevant to cite studies on new particle

We have modified this sentence to be:

It is evident from Figure 1 that even small amounts of $SO_2$ affect the new particle formation substantially, as observed in previous studies (Chu et al., 2016; Liu et al., 2016).

Page 7 line 20:
Before: "Condensation onto existing aerosol particles was prior to the occurrence of new particle formation."
Changed to: "New particles were formed by vapor condensation onto existing aerosol particles."
The new sentence does not make sense. New particles are never formed by growth of existing particles! After reading the text several times I now understand that you mean that condensation was favored compared to new particle formation in the system. Please rewrite to explain your statement correctly.

We have rewritten this sentence.

As long as there was enough seed particle surface area, vapor condensation onto existing aerosol particles was favored compared to the formation of new particles, and this condensation would be the main contribution to the increase of SOA mass.

Page 9 line 20. The authors have added this sentence: "It was also found that during the Investigation of Sulfur Chemistry in the Antarctic Troposphere, the OH tends to increase when the influx of SO2 from above decreases (Mauldin et al., 2004), which means that there is a negative correlation between OH and SO2 in real atmosphere."
It not clear how a study in Antarctica has any relevance to the conditions of the present laboratory study on cyclohexene and SO2 in a high NOx environment.

We agree that this statement does not directly address the issue raised in the previous report. Hence, we deleted it and solely consider our previous first suggested response to this comment, which is below:

Moreover, in real atmospheric situations where $O_3$ is found in much higher proportion than OH, cyclohexene would mainly react with $O_3$ to produce Criegee intermediates, which are good $SO_2$ oxidizers, and significantly less SOA than in the chamber will be formed.

Page 12 line 6-13: My previous question regarding composition and response between samples was only partly answered, so I will try to explain it more specifically: Did the "concentration" (or signal intensity) of organosulfates increase with SO2 concentration?

Currently the authors show a normalized mass spectrum, but I would like to know the variation in intensity.

More HR-MS analyses were performed, with 0 ppb and 236 ppb initial $SO_2$ concentrations (see Figure S6). As can be seen from this figure, the SOA composition changes with initial $SO_2$ concentrations. The NL of the HR-MS results is 2.71E5, 2.28E5 and 1.78E5 when the $SO_2$ concentration is 236, 94 and 0 ppb, respectively. Since a liquid chromatography for quantitative analysis was not combined with the HR-MS in our study, all the HR-MS results were obtained through a direct injection. Combined with the FTIR spectra results, we believe the amount of organosulfates in SOA was increased with initial $SO_2$ concentration.

[Figure]

Figure S6: Comparison of SOA ESI-HR-MS spectra with different initial $SO_2$ concentrations.

We added the following at page 12 line 13.

Comparing HR-MS data when initial concentrations were 0 ppb and 236 ppb reveals that the bands representing organosulfates do not appear 0 ppb of SO2. Peaks at m/z larger than 150 were undetectable at initial $SO_2$ concentration of 0 ppb, while products without sulfur peaked at both concentrations, with the only difference being their intensities. This implies that the process of SOA formation strongly depends on initial $SO_2$ concentrations.

Additional comments:
Please correct the figure caption for Figure 3 to describe the current version.

We have fixed it.

We have revised our manuscript according to the suggestions of the Referee's comments. For clarity, the Referee's comments are reproduced in blue, authors' responses are in black and changes in the manuscript are in red color text. Pages and lines of modified/inserted/deleted texts are relative to the previous version of the manuscript.

**Report #3**

In this study the authors report the formation of organosulfates from the photooxidation of cyclohexene. The manuscript went through the ACPD open discussion process and got mixed reviews, with a number of good suggestions from the more critical reviewers. Based on my examination of the authors' responses to the reviewers' comments, I believe the concerns of the reviewers have been mainly addressed. However, the authors should (1) better discuss the oxidation regime in their experiments and (2) fix the different errors present in the revised version.

To solve the discussion about the impact of ozone in the oxidation of cyclohexene, the authors need to run a simple box model to estimate the degradation pathways. As they are, the explanations are not convincing. Indeed, the authors claim the ozonolysis doesn't contribute to depletion of cyclohexene but the authors didn't propose any estimation of the concentration of OH radicals. It is also surprising that the addition of large quantities of $SO_2$ does not impact more the decay of cyclohexene (e.g. no difference between 40 and 90 ppb).
Finally that's not true that the ozonolysis of cyclohexene produces sCI (c.f. Donahue et al., 2011).
The paragraph 3.2 should be rewritten to discuss the results from a more objective scientific points.

We estimate the degradation pathways of cyclohexene according to the changes of $O_3$ and cyclohexene concentrations, and the rate constants of both $O_3$+cyclohexene and OH·+cyclohexene reactions. About 60% of the total cyclohexene was oxidized by OH, while another fraction of cyclohexene oxidation is attributed to the reaction with $O_3$.
The addition of large quantities of $SO_2$ does not lead to a remarkable cyclohexene decrease. This might be due to the fact that the $SO_2$+OH· reaction quickly reaches saturation, so that additional $SO_2$ does not lead an obvious increase in the consumption of OH. As shown in Figure 1, the particle number concentrations were practically maintained steady when the $SO_2$ concentrations were varied systematically between 30 and 105 ppb. Nucleation was strongly dependent on the abundance of $H_2SO_4$, which was based on the oxidized $SO_2$ + OH radicals (Sihto et al., 2006; Xiao et al., 2015). This feature may indicate that no more sulfates were formed when $SO_2$ concentration was in large excess (>30 ppb) and the OH radicals being insufficient. Higher initial $SO_2$ concentrations may neither decrease the OH concentration substantially, nor induce important cyclohexene decay.
Concerning the sCI formation from cyclohexene-ozone system, Donahue et al.

pointed out that "although we see no evidence of SCI formation from cyclohexene ozonolysis, secondary ozonides have been reported from this system, suggesting SCI formation" (Drozd and Donahue, 2011). The quantitative measure of the sCI yield from cyclohexene ozonolysis was found to be very low, about 3% (Hatakeyama et al., 1984). This is because the Criegee intermediates from endocyclic alkenes are formed with more energy release than those from exocyclic alkenes and hence, are less likely to be stabilized (Chuong et al., 2004). We have revised the discussion about sCI.

The revised part in paragraph 3.2 is as follows.

[revised manuscript text omitted]

---

## Author Response (AR3)

We have revised our manuscript according to the suggestions of the Editor/Reviewers' comments and the responses to the comments are as following. For clarity, the Editor/Reviewer' comments are reproduced in blue, replies to comments are in black and inserted/modified text are in red color text.

Remaining Comments from Reviewer 3:

I suggest that the authors go through the paper, paragraph-by-paragraph, and correct mistakes that bring confusion.

We have gone through the manuscript and corrected several mistakes.

As an example: Even though they have mentioned in their responses that "We have checked and updated the detection limit (0.005 μg mL-1)." the detection limit mentioned in the manuscript is 0.005 mg ml-1...

The detection limit has been corrected in the manuscript to 0.005 $\mu g\ mL^{-1}$.

Overall, I enjoyed reading this paper more this time and the quality has improved but I was hoping for more from the authors.

For instance, the authors should consider identifying the IR bands using the existing literature (e.g. 1040 cm-1, 1195 cm-1; Fig S4 and text).

We have re-written the sentences on Pages 8, Line 35 to Page 9, Lines 1-5 as:
When studying the effect of acidic seed on the growth of isoprene- and α-pinene-based SOA, it was shown that FTIR peaks at 1180 $cm^{-1}$ (C-O-C stretch of hemiacetals and acetals) and 1050 $cm^{-1}$ (C-C-O asymmetric stretch of alcohols) are indicators of acid-catalyzed heterogeneous reactions since these peaks could not, otherwise, be observed in non-acidic conditions (Jang et al., 2002; Czoschke et al., 2003). These peaks are prominent in IR spectra from SOA formation in an acidic particle environment. In the current study, similar peaks were observed at 1195 and 1040 $cm^{-1}$ (see Figure S4).

The relative decays (Fig. S5) don't help much since the initial concentrations varied from 472 to 656 ppb (25-30%!)...The authors should compare the absolute decays and choose experiments when the initial concentrations of cyclohexene are within the same range < 5%.

In a new plot of Figure S5 (below), we have inserted a plot for the amount of reacted cyclohexene with time. Initial concentrations of cyclohexene were 543 and 596 ppb for 0 and 90 ppb of $SO_2$, respectively.

[Figure]

Figure S5: Change of cyclohexene concentration with time at different initial $SO_2$ concentrations. The inner plot shows the amount of consumed cyclohexene with time.

The sentences on Page 9, lines 18-21 were modified as:
It is seen that in the first half hour, the amount of cyclohexene consumed is almost similar for different $SO_2$ concentrations. Regarding the difference between initial cyclohexene concentrations, the similar amount of reacted cyclohexene in the first half hour indicates that low and high OH concentrations were used at high and low $SO_2$ conditions, respectively.

Page 12, Lines 6-12: To validate this paragraph the authors should plot the sum of the organosulfates (area of the peaks) vs the concentrations of the sulfate measured by IC. Once again the relative contributions do not support what the authors are trying to demonstrate (Fig. S6)

The aim of this study was to prove the formation of organosulfates from cyclohexene. While we believe, as the Reviewer suggests, that it is important to quantify these organosulfates, this will be investigated in further studies.

To highlight this, the text on Page 12, Lines 10-14 was modified as:
However, Minerath et al. and Hatch et al. observed an increase in organosulfate yields with increasing sulfate concentration, and sulfate can be regarded as a key parameter influencing the formation of organosulfates (Minerath and Elrod, 2009; Hatch et al., 2011). Since sulfate is formed as a result of $SO_2$ oxidation in the current study, quantification of organosulfates formed from cyclohexene photooxidation will be investigated in further studies in order to examine the effect of increasing $SO_2$ concentration on organosulfates formation.

1.) Please add text in either the discussion section or the conclusions that further work is needed to quantify the abundance of these organosulfates as well as to determine the precursors to the formation of organosulfates. It is likely these gas-phase precursors formed from cyclohexene oxidation also help to explain the acid-catalyzed enhancements in the SOA mass.

We inserted the following text in the manuscript.
Page 12, Line 29-30:
However, quantification of these organosulfates and precursors to their formation should be determined in further studies.

2.) Abstract, Line 19: Please change "first" to "new." Organosulfates being observed is new, but saying first seems like this is the first time ever that organosulfates have been observed in SOA.

This was changed.

3.) Introduction, Page 1, Line 28: change "VOCs oxidation" to "the oxidation of VOCs"

This was changed.

4.) Introduction, Page 2 Line 18: change "area" to "areas"

This was changed.

5.) Introduction, Page 2, Line 21: change "VOCs oxidation" to "oxidation of VOCs"

This was changed.

6.) Introduction, Page 2, Line 25: change "chamber experiments" to "laboratory chamber"

This was changed.

7.) Section Heading 2.3, Page 5: Change "2.3 Products composition analysis" to "2.3 SOA composition analysis"

This was changed.

8.) Section 2.3, Page 5, Line 31: Can you clarify which model of ESI-HR-MS you used and as to whether you mean this was a highi-resolution quadropole time-of-flight

mass spectrometer (ESI-HR-QTOFMS)?

We used an Exactive-Orbitrap mass spectrometer equipped with electro-spray interface (ESI) (Thermo Fisher Scientific, USA), as clarified in the revised manuscript. We have also updated the relevant places in the text.

9.) Section 3.3., Page 12, Line 8: change "organosulfates yield" to "organosulfate yields"

This was changed.

10.) Section 4, Page 12, Line 19: Change "chamber experiments" to "laboratory chamber"

This was changed.

11.) Section 4, Page 12, Line 19: change "secondary aerosols" to "SOA"

This was changed.

12.) Table 2 and elsewhere: Whenever referring to "m/z" please make sure this is italicized.

This was checked and corrected throughout the whole manuscript.

13.) Finally, there have been a number of published field studies that have utilized ESI-MS techniques to characterize urban fine aerosols for orgnaosulfates. From Table 2, have the authors carefully reviewed the literature, especially from urban polluted areas where cyclohexene may be important, to see if these products can be detected? Riva et al. (2015, ES&T), Riva et al. (2016, ACP), and studies by the groups of Sergey Nizordov's (UC-Irvine), Alex Laskin's (PNNL and now Purdue University), Marianne Glasius (Aarhus University), and Jian Zhen Yu (Hong Kong University of Science and Technology) have measured many organosulfates in urban areas that are not due to biogenic VOCs like isoprene and monoterpenes.

We have carefully explored previous studies, including those suggested by the Editor, that have measured organosulfates in urban areas. Few organosulfates with unknown sources, having similar chemical formulae as reported here, have been measured by Hansen et al. 2014 (*Atmos. Chem. Phys.* **2014**, *14*, 7807-7823).

The following was inserted in the manuscript on Page 11, Lines 26-28.
This organosulfate, together with organosulfates with m/z = 179.00181 and 209.01257 measured in this study, were also measured in the Arctic sites, however, with unknown sources (Hansen et al., 2014). This study further supports the formation of

organosulfates from cyclohexene in the atmosphere.

I'm not saying it is required, but it would certainly add more atmospheric implications to your lab study if you could cite papers that have already been published summarizing organosuflate data from urban areas. I think some of these prior studies may measured organosulfates with unknown sources but with similar chemical formulas as you report in Table 2.

[revised manuscript text omitted]